# Disruption of *Plasmodium falciparum* kinetochore proteins destabilises the nexus between the centrosome equivalent and the mitotic apparatus

Jiahong Li [1,2], Gerald J. Shami [3], Benjamin Liffner [4], Ellie Cho [5], Filip Braet[3], Manoj T. Duraisingh [2], Sabrina Absalon[4], Matthew W. A. Dixon [6,7,8] ✉ & Leann Tilley [1,8] ✉

*Plasmodium falciparum* is the causative agent of malaria and remains a pathogen of global importance. Asexual blood stage replication, via a process called schizogony, is an important target for the development of new anti-malarials. Here we use ultrastructure-expansion microscopy to probe the organisation of the chromosome-capturing kinetochores in relation to the mitotic spindle, the centriolar plaque, the centromeres and the apical organelles during schizont development. Conditional disruption of the kinetochore components, *Pf*NDC80 and *Pf*Nuf2, is associated with aberrant mitotic spindle organisation, disruption of the centromere marker, CENH3 and impaired karyokinesis. Surprisingly, kinetochore disruption also leads to disengagement of the centrosome equivalent from the nuclear envelope. Severing the connection between the nucleus and the apical complex leads to the formation of merozoites lacking nuclei. Here, we show that correct assembly of the kinetochore/spindle complex plays a previously unrecognised role in positioning the nascent apical complex in developing *P. falciparum* merozoites.

Malaria is a mosquito-borne disease that causes devastating health problems. Despite substantial efforts over the last decade to reduce the burden of malaria, this disease still caused an estimated 247 million infections, and claimed 619,000 lives, in 2021[1]. The emergence of malaria parasites resistant to the front-line artemisinin-based antimalarials[2], combined with the recent impacts of the COVID-19 pandemic on malaria control efforts[3], makes the need to develop new antimalarial strategies even more pressing.

The process of mitosis in *P. falciparum* differs from that of mammalian cells. Apicomplexan parasites undergo closed mitosis, similar to many fungi, where the nuclear membrane remains intact through the entirety of mitosis[4,5]. During the intraerythrocytic asexual stage, malaria parasites replicate by a process called schizogony, where several asynchronous rounds of mitosis and karyokinesis (nuclear division) are followed by one round of synchronised karyokinesis and cytokinesis to form a segmented

[1]Department of Biochemistry and Pharmacology, Bio21 Molecular Science and Biotechnology Institute, The University of Melbourne, Parkville, VIC, Australia. [2]Department of Immunology and Infectious Diseases, Harvard T. H. Chan School of Public Health, Boston, MA, USA. [3]School of Medical Sciences (Molecular and Cellular Biomedicine) & Australian Centre for Microscopy and Microanalysis, The University of Sydney, Sydney, NSW, Australia. [4]Department of Pharmacology and Toxicology, Indiana University School of Medicine, Indianapolis, IN, USA. [5]Biological Optical Microscopy Platform, The University of Melbourne, Parkville, VIC, Australia. [6]Department of Infectious Diseases, The Peter Doherty Institute, The University of Melbourne, Parkville, VIC, Australia. [7]Walter and Eliza Hall Institute, Parkville, VIC, Australia. [8]These authors contributed equally: Matthew W. A. Dixon, Leann Tilley. ✉e-mail: matthew.dixon@unimelb.edu.au; ltilley@unimelb.edu.au

schizont, resulting in the formation of up to 32 daughter merozoites[5,6].

In many eukaryotic cells, the centrosome is the microtubule organising centre (MTOC) for the chromatid-separating spindle microtubules. In mammalian cells, the centrosome contains a pair of centrioles, surrounded by pericentriolar material. During mitosis, the centrosome duplicates and the nuclear membrane is disassembled[7]. Mitotic spindle microtubules are nucleated from the centrosome and extend toward the centre of the cell to capture structures called kinetochores, which, in turn, segregate sister chromatids[8].

By contrast, *P. falciparum* has no canonical centrioles, nor any other distinct structure within the centrosome equivalent, which is termed the centriolar plaque. Early electron microscopy (EM) studies revealed an amorphous structure embedded in the nuclear membrane, with electron-dense signals detected on both sides of the membrane[9–12]. The molecular composition and architecture of the *Plasmodium* centriolar plaque remain poorly understood. A limited number of canonical centrosome components have been identified. These include centrins−structural proteins that belong to a small family of calcium-binding proteins with four EF-hand domains[13–17]. A Sfi1-like protein (*Pf*Slp)[18] and a homologue of the Aurora kinase (*Pf*Ark1)[19] have also been identified. Each of these proteins is associated with the outer centriolar plaque region.

3D-reconstruction electron microscopy of the mitotic spindle of *P. falciparum* schizonts revealed 14 pairs of kinetochores[12]. Many kinetochore proteins have been determined in *P. berghei* and the related apicomplexan parasite, *Toxoplasma gondii*[20–23]. A key component of the kinetochore is a tetrameric complex that comprises heterodimers of Nuclear Division Cycle 80 protein (NDC80) and Nuf2, with heterodimers of Spindle Pole Component (SPC)-24 and SPC25. The C-termini of the SPC24/SPC25 heterodimer provide a link to CENH3 (CENP-A) a component of the centromeres, while the NDC80-Nuf2 heterodimer interacts with the spindle microtubules[22,24–26].

In mammalian cells, CENP-A depletion leads to destabilisation of the kinetochore complex, which, in turn, compromises spindle microtubule stability, resulting in release of microtubule minus-ends from the centrosome and dispersal of the pericentriolar material[27]. These data suggest a functional interdependence of the centromeres and the centrosomes during mitosis[27].

NDC80 and Nuf2 have been studied in *P. berghei* blood stages using live cell reporters, and are first detected in the trophozoite stage, as mitosis begins[23,28]. In segmented schizonts, which have completed mitosis, the NDC80 signal is still detected, while the Nuf2 signal is not[23,28]. Previous studies showed that NDC80 can distribute along the nuclear microtubules in *Plasmodium*[17,20], in contrast to mammalian kinetochores, which bind predominantly at the plus ends. Efforts to knockout or knockdown NDC80 and Nuf2 in *P. berghei* were not successful, consistent with these genes being essential[20,23]. Similarly, knock sideways and promoter swap methods in *P. falciparum* and *P. yoelii* confirm that NDC80 plays an essential role in replication[28–30].

In this work, we generate transfectants expressing tagged versions of the kinetochore proteins, *Pf*NDC80 and *Pf*Nuf2, under conditional knockout/knockdown control. Combining classical cell and molecular biology techniques with live cell imaging, Ultrastructure-Expansion Microscopy (U-ExM) and electron microscopy, we demonstrate that *Pf*NDC80 and *Pf*Nuf2 play essential and unexpected roles in asexual blood stage proliferation. Deletion of *Pf*NDC80 and *Pf*Nuf2 leads to loss of the signal for the centromere marker, CENH3, suggesting disruption of the centromere complex. It also breaks the nexus between the spindle microtubules and the nuclear membrane-embedded centriolar plaque/MTOC, leading to disorganised spindles and uncoupling of the nucleus from the apical complex. The centrin-containing outer centriolar plaque disengages from the nucleus, but still directs assembly of sub-pellicular microtubules and apical organelles, leading to packaging of anucleate merozoites.

## Results

### *Pf*NDC80 and *Pf*Nuf2 undergo dynamic reorganisation

We targeted the native locus of the kinetochore components, *Pf*NDC80 (PF3D7_0616200) and *Pf*Nuf2 (PF3D7_0316500), using a CRISPR-Cas9 homology driven approach, incorporating a 3×HA tag and a *glmS* ribozyme sequence into the 3′ end of the loci. *LoxP* sites were introduced downstream of the HA tag via the insertion of an artificial intron within the coding region (Supplementary Fig. 1a). Introduction of *loxP* and *glmS* ribozyme sequences allows for conditional knockout (cKO) of the genes and knockdown (cKD) of the transcripts, respectively[31,32] (Supplementary Fig. 1a), which ensures very efficient and sustained knockdown of the gene product. Integration into the endogenous loci and excision progress of target gene were monitored at different time points by PCR (Supplementary Fig. 1b, c). The level of expression of the HA-tagged proteins was monitored by Western blotting (Supplementary Fig. 1d, e). Wide-field immunofluorescence microscopy revealed *Pf*NDC80 and *Pf*Nuf2 as punctate structures within the nuclei (Supplementary Fig. 1f, g).

Following validation of the cell lines, we examined the location of the two kinetochore components at different stages of mitosis (Fig. 1a). U-ExM has recently been applied to visualise a range of cellular processes in *P. falciparum*[16,33–37]. Here, the *Pf*NDC80 and *Pf*Nuf2 transfectant-infected red blood cells (RBCs) were embedded, and gels were expanded 4.5 times (on average), labelled, mounted, and imaged by Airyscan confocal microscopy. NHS ester-conjugated, sulfonated fluorophores are water-soluble probes that react with primary amines in proteins and other molecules to yield stable amide bonds. Suspending DyLight™ 488-NHS ester in phosphate-buffered saline (PBS) (NHS-PBS) gives good definition of protein-rich structures, such as the rhoptries[37] (Fig. 1b, right-hand side, depicted in greyscale/inverse greyscale). Here, we explored a different method for general protein labelling, which yields improved membrane labelling. We found that applying DyLight™ 488-NHS ester suspended in 3% bovine serum albumin (NHS-BSA) gives good delineation of the RBC membrane, the parasite plasma membrane (PPM) and nuclear membrane (Fig. 1b, left-hand side, depicted in greyscale/green). We used both protocols in this study, with the NHS-PBS fluorescence signal depicted in inverted greyscale and the NHS-BSA signal in green. The HA-tagged *Pf*NDC80 and *Pf*Nuf2 structures were visualised using anti-HA antibodies. Progress through blood stage mitosis is readily tracked by labelling microtubule structures with anti-β-tubulin or anti-α-tubulin and the chromatin with DAPI.

The dynamic organisation of kinetochore proteins, Nuf2, NDC80 and Apicomplexan Kinetochore protein1 (AKiT1), has previously been studied in *P. berghei*[5,20,23,38,39]. Consistent with those reports, in nuclei preparing for karyokinesis, the *Pf*NDC80 and *Pf*Nuf2 signals (Fig. 1c and Supplementary Figs. 2a, b, 3a, b, first rows, yellow arrowheads) are concentrated near the base of anti-β-tubulin-labelled hemispindles, i.e., at the periphery of the nucleus, as marked by DAPI (blue) and the NHS-BSA-labelled membrane (green) (Fig. 1c and Supplementary Figs. 2a, 3a). During nuclear division, *Pf*NDC80 and *Pf*Nuf2 signals (Fig. 1c and Supplementary Figs. 2a, b, 3a, b, second row, yellow arrowheads) are often observed as a pair of parallel lines at the midplane of the mitotic spindle. In some cells, extended spindles, stretching between separating nuclei, are observed (Fig. 1c, second row, zoom 1, yellow arrowheads), with the kinetochore marker concentrated towards the spindle origins. In segmenting schizonts (Fig. 1c and Supplementary Figs. 2a, b, 3a, b, third row), *Pf*NDC80 and *Pf*Nuf2 (yellow arrowheads) are associated with the contracted spindle remnant. Nascent sub-pellicular microtubules (white arrowheads) are evident just above the spindle remnant, as reported previously[5,38,39]. In segmented schizonts (Fig. 1c and Supplementary Figs. 2a, b, 3a, b, fourth row), the kinetochore components (yellow arrowheads) contract back to a location at the nuclear periphery, and the sub-pellicular microtubules (white arrowheads) are evident outside the nucleus

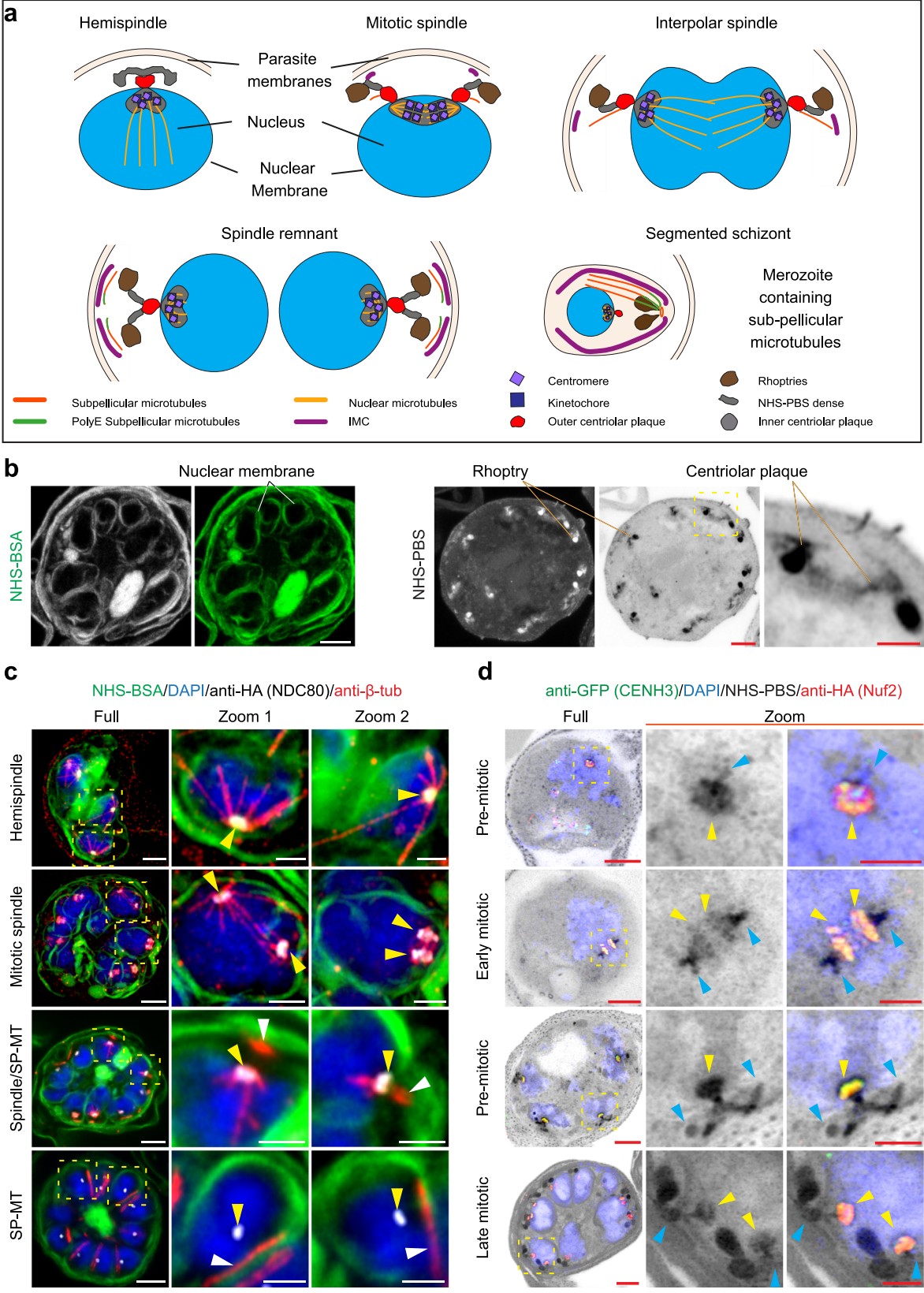

under the PPM, consistent with previous reports[16,20,23,35]. The NHS-PBS counterstained samples (Supplementary Figs. 2b, 3b), reveal that the kinetochore marker is associated with protein-rich structures that migrate from the nuclear periphery to the spindle midplane and back again.

## Kinetochore and centromere markers are co-located during nuclear division

To investigate the centromere location relative to *Pf*NDC80 and *Pf*Nuf2, we used a previously generated a GFP-*Pf*CENH3 (PF3D7_1333700) reporter construct[17] and transfected it into the

**Fig. 1 | *Pf*NDC80-HA and *Pf*Nuf2-HA localisation in *P. falciparum* asexual blood stages. a** Illustration of *P. falciparum* mitotic stages. The diagrams represent parasites with a hemispindle, a mitotic spindle, an interpolar spindle, a spindle remnant and a formed merozoite. The apical complex is assembled above a nuclear membrane-embedded centriolar plaque, with inner and outer regions. Structures are labelled in the diagrams or defined in the key at the bottom of the diagram. **b** U-ExM images of schizont-infected RBCs labelled with DyLight488-NHS-ester in BSA (NHS-BSA, greyscale and green) or DyLight488-NHS-ester in PBS (NHS-PBS, greyscale, and inverted greyscale). NHS-BSA (left panels) gives good labelling of membranes. The nuclear membranes are highlighted. NHS-PBS (right panels; with zoom inset) gives good labelling of protein-rich structures, such as the rhoptries and centriolar plaque. U-ExM images of (**c**) *Pf*NDC80-HA transfectants (anti-HA (*Pf*NDC80), greyscale, yellow arrowheads), counter-stained with NHS-BSA (green)

and (**d**) *Pf*Nuf2-HA/GFP-*Pf*CENH3 transfectants (anti-HA (*Pf*Nuf2), green, and anti-GFP (*Pf*CENH3); red, yellow arrowheads) counter-stained with NHS-PBS (greyscale), revealing the kinetochore and centromere components co-located and relocating from the base of the hemispindles to the mid-plane of the mitotic spindle, then contracting to the nuclear periphery. The images are displayed as z-projections. Anti-β-tubulin (anti-β-tub, red) marks the spindle microtubules (intranuclear) and sub-pellicular microtubules (SP-MT, white arrowheads). DAPI (blue) stains the chromatin. Zoom-out/zoom-in scale bars are 5 μm and 2 μm. Additional images/channels are presented in Supplementary Fig. 2 (*Pf*NDC80-HA), Supplementary Fig. 3 (*Pf*Nuf2-HA) and Supplementary Fig. 5a (*Pf*Nuf2-HA/ GFP-*Pf*CENH3) and 5b (*Pf*NDC80-HA/ GFP- *Pf*CENH3). Figure 1c, d experiments were performed 3 times from 3 different sample preparations.

*Pf*NDC80-HA and *Pf*Nuf2-HA lines (Supplementary Fig. 4). Figure 1d illustrates the co-location of the centromere (*Pf*CENH3, green) and kinetochore (*Pf*Nuf2, red) markers. Prior to nuclear division, the kinetochores and centromeres are sequestered to the strongly labelled NHS-PBS inner centriolar plaque region (Fig. 1d and Supplementary Fig. 5a, top row, yellow arrowhead). As the centriolar plaque duplicates and separates, the kinetochore (*Pf*Nuf2, red) and centromere (*Pf*CENH3, green) markers migrate in a pattern that overlaps with the NHS-PBS bright material (Fig. 1d and Supplementary Fig. 5a, second row, yellow arrowheads). An equivalent pattern is observed for *Pf*NDC80-HA and GFP-*Pf*CENH3 (Supplementary Fig. 5b.) The data suggest that the NHS-PBS bright, inner centriolar plaque is comprised of sequestered kinetochore and centromere complexes.

### Reorganisation of the centriolar plaque, kinetochore and apical complex

We used a monoclonal anti-centrin antibody to probe the reorganisation of the centriolar plaque at different stages of nuclear division. Early in division the centrin-labelled punctum is located on the cytoplasmic side of the nuclear membrane (Supplementary Fig. 6a; yellow arrowheads), directly opposite the site of formation of the hemispindle (green microtubules) (Supplementary Fig. 6a). Following completion of nuclear division, the spindle microtubules contract to stubs (green signal, aqua arrowheads) that sits just below the centrin punctum (yellow arrowheads) (Supplementary Fig. 6c). At this stage, short stubs of cytoplasmic microtubules (white arrowheads) are observed, apparently arising from the centrin-labelled punctum. Additional examples of each step described above are provided in Supplementary Fig. 7a–c. These data are consistent with previously published data[37].

Developing rhoptries are evident as very strongly NHS-PBS-labelled bulbous structures located towards the outside of the cell (Supplementary Figs. 6b-d, 7b, c, 8 purple arrowheads). In earlier rounds of nuclear division, each nucleus is associated with one rhoptry (Supplementary Figs. 6b, 7b, first and second row), increasing to two rhoptries in the final round of nuclear division (Supplementary Figs. 6c, 7b, third and fourth row). A continuum of NHS-PBS-labelling is observed from the rhoptries to the centrin-labelled puncta that lie at the surface of the nucleus (Supplementary Figs. 6b-d, 7b, 8), as reported previously[17,35].

In NHS-BSA-labelled schizonts that are approaching maturity, anti-β-tubulin-labelled sub-pellicular microtubules (Supplementary Fig. 6e, aqua arrowheads) are evident outside the nucleus, directly above the puncta of retracted spindle microtubules (Supplementary Fig. 6e, yellow arrowheads). Subpellicular microtubules are partially polyglutamylated, and one end of these apically located microtubule structures is recognised by anti-polyE antisera, confirming they are subpellicular microtubules (Supplementary Fig. 6e, orange arrowheads), as previously described[17,33]. Anti-*Pf*GAP45 (greyscale), a marker of the inner membrane complex (IMC)[40], is evident as a cap above each nucleus (Supplementary Fig. 6f, magenta arrowheads). The sub-

pellicular microtubules (aqua arrowheads) lie under the IMC, close to the intra-nuclear spindle remnants (Supplementary Fig. 6f, yellow arrowheads). Additional examples are presented in Supplementary Fig. 9a (anti-polyE) and Supplementary Fig. 9b (anti-*Pf*GAP45).

Taken together with previous reports[17,35], these data indicate connections from the spindle and inner centriolar plaque through to the outer centriolar plaque, the nascent cytoplasmic microtubules, the apical complex, and the IMC and the PPM. The data show that the spindle microtubules are initiated from a protein-dense structure inside the nucleus, that contains the kinetochore proteins, *Pf*Nuf2 and *Pf*NDC80. In addition, we show that cytoplasmic microtubules nucleate close to the outer centrin-containing plaque, corresponding to either the centriolar plaque itself or a closely apposed nascent apical polar ring.

### *Pf*NDC80 deletion impacts spindle microtubules and centromeres

We subjected *Pf*NDC80-HA-*diCRE-glmS* transfectants to rapamycin/glucosamine treatment for 41 h (initiated at 0–2 h post-invasion (hpi)). This dual conditional knockout and knockdown approach is abbreviated as cKO/KD. PCR and Western analysis show excision and almost complete loss of protein by 30 hpi (Supplementary Fig. 1b, d). As anticipated, cKO/KD of *Pf*NDC80 severely impacts the survival of the blood stage parasites with parasites unable to complete schizogony (Fig. 2a). To investigate the defect more closely, we examined the physical organisation of the mitotic machinery in tightly synchronised early- and mid-stage mitotic schizonts (at 30 and 36 hpi), labelled with anti-β-tubulin and DAPI and imaged using U-ExM (Fig. 2b, c). In control cells, hemispindles, mitotic spindles and extended spindles are observed (Fig. 2b, c, control, yellow arrowheads). In *Pf*NDC80-cKO/KD transfectants, the spindle microtubules show a disrupted organisation. In early-stage cells with a single nucleus (Fig. 2b, cKO/KD, fourth row, yellow arrowheads), elongated microtubules are evident that appear to have been separated from the bulk nuclear material. As nuclear division progresses, microtubules continue to form in the nucleus but fail to converge at an MTOC at the nuclear periphery (Fig. 2b, cKO/KD, fourth and sixth row, yellow arrowheads); and typical hemispindle and mitotic spindle structures are absent (Fig. 2b, c, cKO/KD, yellow arrowheads). Abnormal microtubules were observed in all anti-HA-negative *Pf*NDC80-HA cKO/KD cells examined at 36 and 38 hpi (Fig. 2d).

We next examined the reorganisation of the centromere marker, *Pf*CENH3, following cKO/KD of *Pf*NDC80. As described above, in control cells *Pf*NDC80-HA (red) and GFP-*Pf*CENH3 (green) are co-located during chromatid duplication and separation (Fig. 3a, yellow arrowheads, control). As expected, disruption of *Pf*NDC80 leads to loss of the red fluorescence signal (Fig. 3a, cKO/KD). Interestingly, the signal for GFP-*Pf*CENH3 (green) also disappeared in anti-HA-negative *Pf*NDC80-HA/GFP-*Pf*CENH3 cells (Fig. 3a, b, cKO/KD). This suggests that sequestration of *Pf*CENH3 into kinetochore/centromere complexes is lost and that *Pf*CENH3 either disperses throughout the

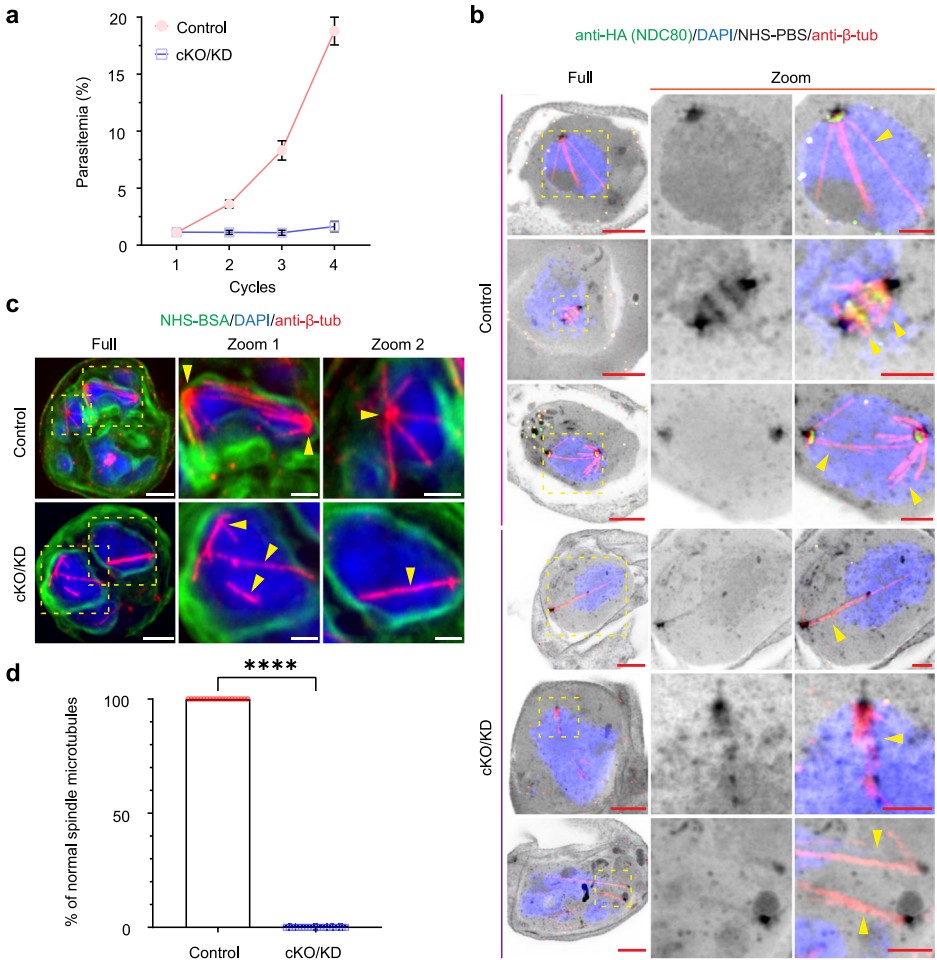

**Fig. 2 | *Pf*NDC80 is required for parasite growth and is required for correct spindle microtubule positioning. a** Assay of asexual growth of *Pf*NDC80-HA parasites following conditional knockout/knockdown (cKO/KD). Parasites were treated with rapamycin and glucosamine (cKO/KD) or DMSO (control) and the parasitemia was monitored for 4 cycles by flow cytometry. The experiment was performed 3 times, with means and standard deviations presented. Representative U-ExM images of control and *Pf*NDC80-cKO/KD schizonts during early (**b** 30 hpi) and mid (**c** 36 hpi) rounds of nuclear division. *Pf*NDC80-HA is visualised using anti-HA (green, **b**). The spindle microtubules (yellow arrowheads) were labelled with anti-β-tubulin (red, anti-β-tub), chromatin was probed with DAPI (blue), and NHS-PBS (inverse grayscale, **b**) was used to label the general proteins while the NHS-BSA (green, **c**) was used to enhance membrane labelling. The images are displayed as z-projections. All the image scale bars are 5 µm, except the zoom images, which are 2 µm. Figure 2b experiments were performed 1 time and Fig. 2c experiments were performed 3 times. **d** Analysis of the percentage of normal spindle microtubules in anti-HA-positive control and anti-HA-negative cKO/KD *Pf*NDC80-HA cells (36 and 38 hpi). Control, *n* = 20; cKO/KD, *n* = 20. The images were analysed from 3 independent experiments. The mean and standard deviation are plotted. Individual data points are shown. Statistical differences were determined using an unpaired Mann–Whitney *t*-test (****$p < 0.0001$).

nucleus or is unstable and becomes degraded. Individual channels and additional examples of early, mid, late and segmented schizonts (control and cKO/KD) are presented in Supplementary Fig. 10.

## Deletion of *Pf*NDC80 impacts merozoite formation and cytokinesis

We investigated if *Pf*NDC80 cKO/KD affects the final round of nuclear division (38–40 hpi) when the cell initiates cytokinesis and the nuclei are packaged into forming merozoites. In control cells, the spindle remnant (Fig. 3c, control, yellow arrowheads) is positioned towards the outside of the parent cell, close to the centrin punctum (control, orange arrowheads). Upon disruption of *Pf*NDC80, severe disorganisation of the schizont is observed. The spindle microtubule remnant is located away from the nuclear periphery (Fig. 3c, cKO/KD, Zoom 1, yellow arrowheads), while the centrin puncta are aggregated in a region of the cell away from the nuclei (Fig. 3c, cKO/KD, Zoom 2, orange arrowheads). Quantitation reveals that in controls, all the centrin puncta are located just outside of the nuclear membrane, while in contrast only 18% of the centrin puncta are located correctly following cKO/KD of *Pf*NDC80 (Fig. 3d). In control cells, the nascent sub-pellicular microtubules, marked by anti-polyE (Fig. 3e, aqua arrowheads), are evident as narrow structures that are located under the IMC, marked by anti-*Pf*GAP45 (Fig. 3f, magenta arrowheads). In the cKO/KD parasites, the sub-pellicular microtubule/IMC complex still forms and is located close to the centrin puncta, but is distal to the nuclei (Fig. 3c, e, f, cKO/KD). Additional examples and individual channels are presented in Supplementary Fig. 11a, b (anti-centrin), c (anti-polyE), d (anti-*Pf*GAP45).

## *Pf*NDC80 is required for correct schizont segmentation

We next imaged fully segmented schizonts to visualise the impact of *Pf*NDC80 cKO/KD on merozoite formation. In control cells, the NHS-BSA signal reveals fully formed merozoites, each containing a compact nucleus located at the basal end of the merozoite (Fig. 4a, b, control). The *Pf*GAP45-labelled IMC forms a sheath around each of the merozoites in the segmented schizont (Fig. 4a, magenta arrowheads). The polyE labelled sub-pellicular microtubules radiate from the apical end of the merozoite (Fig. 4b, aqua arrowheads).

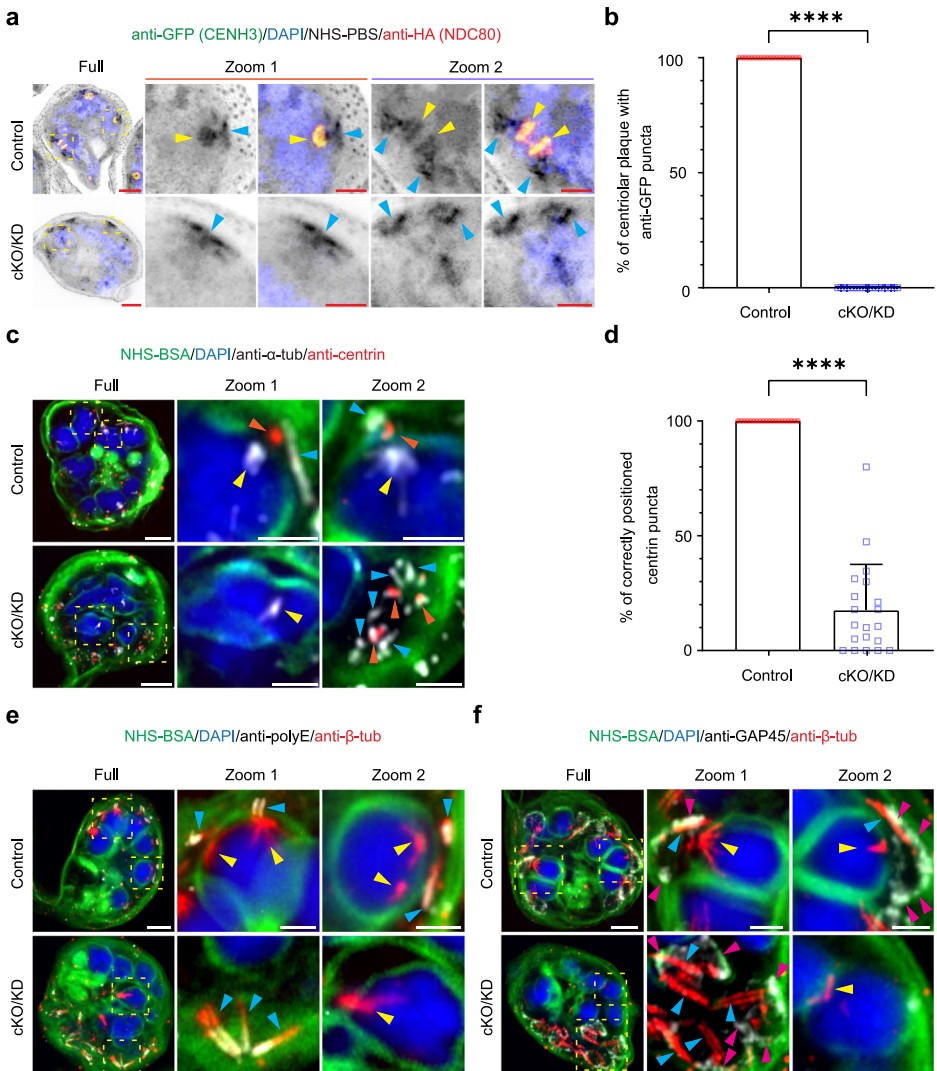

**Fig. 3 | *Pf*NDC80 is required for correct positioning of centromeres, the outer centriolar plaque and the apical complex. a** Representative U-ExM micrographs of *Pf*NDC80-HA/GFP-*Pf*CENH3 control and conditional knockout/knockdown (cKO/KD) mid stage schizonts counterstained with the general protein dye NHS-PBS (inverse grayscale). The anti-HA (*Pf*NDC80-HA, red) labelling overlaps with the anti-GFP (GFP-*Pf*CENH3, green) labelling (yellow arrowheads). In cKO/KD cells, NHS-PBS-labelled apical complex (aqua arrowheads) is separated from the DAPI-labelled nucleus (blue). The anti-GFP signal is also lost. Additional images are presented in Supplementary Fig. 10. The experiment was performed 3 times. **b** Analysis of the percentage of centriolar plaque structures (labelled with NHS-PBS) associating with anti-GFP puncta in anti-HA-positive control and anti-HA-negative cKO/KD *Pf*NDC80-HA/GFP-*Pf*CENH3 cells (34 and 36 hpi). Control, n = 20; cKO/KD, n = 20. The mean and standard deviation are plotted. Individual data points are shown. An unpaired Mann–Whitney *t*-test was performed (****$p < 0.0001$). The images for quantifications were acquired from 3 independent experiments. Figure 3a, b experiments were repeated 3 times. **c** In late stage schizonts (38 hpi), anti-centrin labels the outer centriolar plaque (red, orange arrowheads), while anti-α-

tubulin (greyscale, anti-α-tub) labels the sub-pellicular (aqua arrowheads) and spindle remnant (yellow arrowheads) microtubules. Additional images are presented in Supplementary Fig. 11a, b. The experiment was performed two times. **d** Analysis of centrin puncta positions. Control, n = 20; cKO/KD, n = 20. The mean and standard deviation are plotted. Individual data points are shown. An unpaired Welch's *t*-test was performed to test significance (****$p < 0.0001$). The images for analysis were acquired from 2 independent experiments. Figure 3c, d experiments were repeated 2 times. **e** Late stage mitotic schizonts (38 hpi). The sub-pellicular microtubules (aqua arrowheads) are distinguished by labelling with anti-β-tubulin (red, anti-β-tub) and anti-polyE (greyscale). The experiment was performed one time. **f** The inner membrane complex is labelled with anti-*Pf*GAP45 (greyscale, magenta arrowheads, anti-GAP45). The post-mitotic spindle remnant, inside the nucleus, is labelled with anti-β-tubulin (red, yellow arrowheads). Additional images are presented in Supplementary Fig. 11c, d. The experiment was performed twice. The images are displayed as z-projections. All the image scale bars are 5 μm, except the zoom images, which are 2 μm.

In *Pf*NDC80 cKO/KD parasites, the apical structures, delineated by the IMC marker (*Pf*GAP45) and polyE-labelled subpellicular microtubules, accumulate in one region of the parent cell. The apical complexes are physically separated from the nuclei, which accumulate in another region of the schizont (Fig. 4a, b, cKO/KD, Supplementary Fig. 12a, b, cKO/KD). Quantitative analysis of control and *Pf*NDC80 cKO/KD parasites revealed that 66% of nuclei are unpackaged (i.e., DAPI-positive, *Pf*GAP45-negative) (Fig. 4c), and 58% of merozoites are nucleus-free (i.e., *Pf*GAP45-positive but DAPI-negative) (Fig. 4d,

Supplementary Fig. 12c, d). We also analysed the ratio of the number of merozoites to the number of nuclei per cell. In controls, there were equal numbers of nuclei and apical complexes (i.e., ratio of 1), while in the *Pf*NDC80 cKO/KD parasites the ratio was 0.8, suggesting a partial defect in nuclear division in addition to the nucleus capture defect (Supplementary Fig. 12e). Closer examination of the images suggests inconsistency in the size of the individual nuclei being produced in the *Pf*NDC80-cKO/KD parasites. Quantification reveals a significant decrease in the mean (expanded) volume of the nuclei of cKO/KD

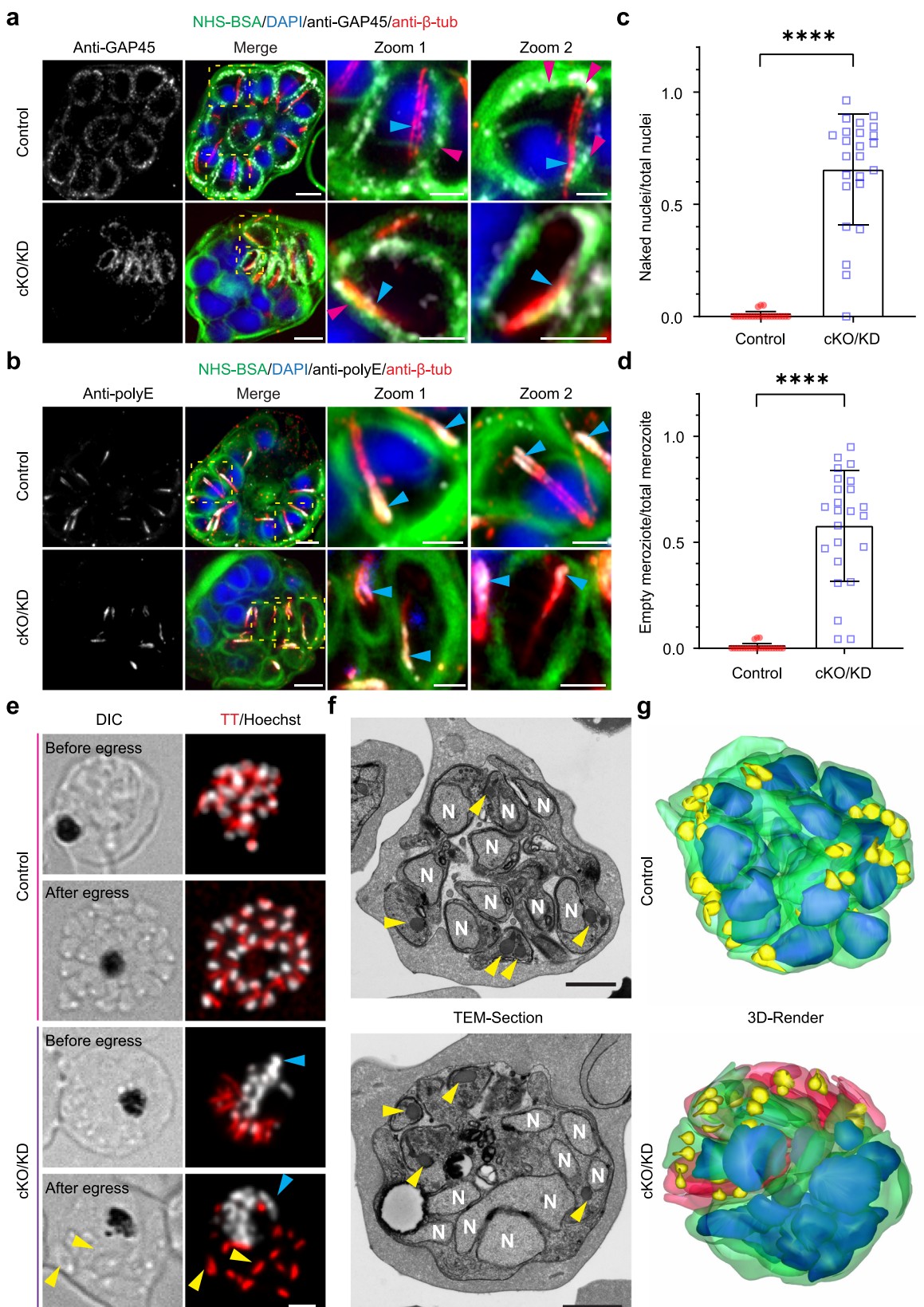

parasites ($37 \pm 34$ µm³) compared with controls ($43 \pm 10$ µm³), as well as the presence of some very large volume nuclei (up to 200 µm³), consistent with uneven inheritance of the nuclear contents (Supplementary Fig. 12f).

Live cell imaging of control *Pf*NDC80-tagged mature schizonts and egressing merozoites (41 hpi) reveals fully segregated nuclei, with Tubulin-Tracker-labelled (TT, red) subpellicular microtubules adjacent to the nuclei (Fig. 4e, control). In *Pf*NDC80-cKO/KD schizonts (Fig. 4e, cKO/KD), the nuclei (aqua arrowheads) are segregated away from the subpellicular microtubules. Interestingly, *Pf*NDC80-cKO/KD schizonts still appear to undergo egress as evidenced by differential interference contrast (DIC) imaging. The daughter merozoites appear to lack nuclei

**Fig. 4 | *Pf*NDC80-deficient parasites fail to form intact merozoites.**
**a**, **b** Representative U-ExM images of segmented schizonts in control (DMSO) and *Pf*NDC80-cKO/KD (conditional knockout/knockdown, treated with rapamycin and glucosamine) parasites. The sub-pellicular microtubules are labelled with anti-β-tubulin (anti-β-tub, red, aqua arrowheads) and the chromatin with DAPI (blue). NHS-BSA (green) counterstains the membranes. **a** Anti-*Pf*GAP45 labels the inner membrane complex (anti-GAP45, greyscale, magenta arrowheads). The experiments were performed twice. **b** Anti-polyE labels the sub-pellicular microtubules at their minus ends (greyscale, aqua arrowheads). The experiment was performed one time. All images are presented as z-projections. All scale bars are 5 μm, except the zoom images which are 2 μm. Additional images are presented in Supplementary Fig. 12a, b. **c**, **d**. Quantification of nuclei parameters for control and *Pf*NDC80-cKO/KD segmented schizonts based on U-ExM data (*n* = 24 cells). The fraction of unpackaged nuclei **c** and akaryotic merozoites **d** are presented (*n* = 24 cells). Statistical differences were determined using an unpaired Welch's *t*-test for **c** and

**d** (****p < 0.0001). Further quantitative analysis is presented in Supplementary Fig. 12c–f. Figure 4c, d images for analysis were acquired from three independent experiments. **e** Live cell imaging of control and *Pf*NDC80-cKO/KD segmented schizonts before (upper row) and after (lower row) egress. The subpellicular microtubules were labelled with Tubulin Tracker (TT, red) and chromatin was labelled with Hoechst (greyscale). The yellow arrowheads indicate akaryotic merozoites and blue arrowheads indicate naked nuclei. All images are presented as z-projections. The scale bar is 2 μm. The experiments were performed 3 times. **f** Single section electron micrographs of segmented schizonts in control and *Pf*NDC80-cKO/KD parasites. The nuclei (N) are marked and the rhoptries are indicated with yellow arrowheads. **g** 3-D array tomography reconstructions of segmented schizonts in control and *Pf*NDC80-cKO/KD parasites. Figure 5f, g experiments were performed one time. Colour legend: Green, nucleated daughter merozoite plasma membrane; red, anucleate daughter merozoite plasma membrane; blue, nuclei; yellow, rhoptries. **f** Scale bar is 1 μm.

(Fig. 4e, yellow arrowheads). Taken together, our results demonstrate that *Pf*NDC80 loss is associated with a defect in segmentation of schizonts.

## Electron microscopy analysis confirms the mislocation of the nuclei upon *Pf*NDC80 cKO/KD

To investigate the ultrastructural morphology of the *Pf*NDC80-cKO/KD parasites at the nanoscale, we performed thin section transmission electron microscopy of segmented schizonts (Fig. 4f). In control cells, newly formed merozoites are evident. The electron dense apical organelles, e.g., rhoptries (yellow arrowheads), are positioned at one end of the merozoite, while the nucleus is observed in the basal region (Fig. 4f, control). This arrangement can be clearly visualised in 3D-rendered Array Tomography data showing the rhoptries (yellow) locating at the apical end and the nuclei (blue) at the basal end of the merozoites (membranes in green) (Fig. 4g, control, see Video S1). In contrast, in the *Pf*NDC80-cKO/KD schizonts, the nuclei are aggregated to one side of the cell. Merozoites that lack nuclei but contain rhoptries (yellow arrows) and regions of IMC (observed as membrane thickening), are clustered on the other side of the schizont (Fig. 4f, cKO/KD). This is readily visualised in the Array Tomography data (Fig. 4g, cKO/KD), and associated 3D rotations (Video S2), where the akaryotic merozoites and nuclei are observed on opposing sides of the schizont. These data confirm the live cell and U-ExM data showing mis-segregation of the nuclei and akaryotic merozoites in *Pf*NDC80-cKO/KD schizonts.

## *Pf*Nuf2 is also essential for normal karyokinesis and cytokinesis

We next investigated the function of a second kinetochore localising protein, *Pf*Nuf2. cKO/KD of the *Pf*Nuf2-HA-*diCRE-glmS* locus was confirmed by PCR and Western blotting, showing an almost complete gene excision and loss of protein by 30 hpi (Supplementary Fig. 1c, e). As anticipated, disruption of *Pf*Nuf2 seriously impacts parasite replication, with a 90% reduction in growth, when compared to the controls (Supplementary Fig. 13a).

*Pf*Nuf2-cKO/KD parasites exhibit a similar phenotype to *Pf*NDC80-cKO/KD parasites. In early nuclear division (30 hpi), NHS-PBS-labelled structures are observed associated with aberrant spindles microtubules or with microtubules that are apparently disconnected from the nucleus (Fig. 5a, cKO/KD, second row, aqua arrowheads, Supplementary Fig. 13b, cKO/KD, yellow arrowheads). In mid mitosis during karyokinesis, dividing nuclei harbouring aberrant or no spindle microtubules are observed (Fig. 5a, fourth row, 5b, cKO/KD; Supplementary Fig. 13c, cKO/KD, yellow arrowheads). In late and segmented schizonts, disconnected sub-pellicular microtubules are observed (Supplementary Fig. 13d, e, cKO/KD, aqua arrowheads). As observed for *Pf*NDC80 cKO/KD, the *Pf*CENH3 signal is lost following cKO/KD of *Pf*NUF2, indicating disruption of the centromeres (Fig. 5c, cKO/KD). Detailed examination of these NHS-PBS-labelled structures in the *Pf*Nuf2-cKO/KD parasites shows that the outer centriolar plaque

and nascent apical complex are still generated but have lost the connection to the nucleus and are instead located close to the PPM (diagram in Fig. 6).

In *Pf*Nuf2-cKO/KD parasites undergoing early (Fig. 5d, second row, Supplementary Fig. 14a, cKO/KD), mid (Fig. 5d, fourth row, Supplementary Fig. 14c, cKO/KD) and late rounds of nuclear division (Supplementary Fig. 14b, d, cKO/KD) and in segmented schizonts (Fig. 5e, cKO/KD), the arrangement of the centrin-labelled puncta (orange arrowheads) is disorganised. In the mid nuclear division, multiple centrin-labelled puncta are observed associated with one nucleus (Supplementary Fig. 14c, cKO/KD, Zoom 1), while other centrin puncta aggregate in regions of the cell distal to the nuclei (cKO/KD, Zoom 2). In later nuclear division, the association of the apical complex, nascent sub-pellicular microtubules (aqua arrowheads) and rhoptries (purple arrowheads) is maintained but the outer centriolar plaque (labelled with anti-centrin, orange arrowheads) is disconnected from the nucleus (Supplementary Fig. 14b, d, cKO/KD).

The subpellicular microtubules are associated with rhoptries (purple arrowheads) and the *Pf*GAP45-labelled IMC (Fig. 5f, magenta arrowheads). The subpellicular microtubules (Fig. 5g, aqua arrowheads) are polyglutamated (anti-polyE, white arrowheads) but are separated from the nuclei (Fig. 5g, cKO/KD). An NHS-PBS-labelled structure that likely represents the remnant spindle and inner centriolar plaque (kinetochores and associated components) is present at the nuclear periphery in the control cells (Fig. 5f, control, yellow arrowheads). Regions located away from the nuclear periphery with a similar staining profile can be observed in the *Pf*Nuf2-cKO/KD parasites but we are unable to confirm the protein composition (Fig. 5f, g, cKO/KD, green arrowhead). Additional data showing different stages of schizogony and individual zoom-in panels, with NHS-PBS labelling are presented in Supplementary Fig. 15a (*Pf*GAP45 labelling) and 15b (polyE labelling). These data confirm that cKO/KD of *Pf*Nuf2 results in a loss of the connection between the intranuclear spindle and the apical organelles, leading to aberrant segregation of the centriolar plaque, sub-pellicular microtubules and IMC away from the nuclei (Diagram in Fig. 6).

## Discussion

Early electron microscopy studies revealed that spindle microtubules are nucleated from a nuclear membrane-spanning structure, referred to as the centriolar plaque, and showed that the spindle is decorated with kinetochores[12,41,42]. Following several rounds of nuclear division, the secretory and invasion machineries of the nascent merozoites are assembled in an apical complex, just above the site of the centriolar plaque. As segmentation proceeds, an apical complex is assembled and a nucleus is captured by each forming merozoite[43]. However, the physical basis for the coordination of these events was unclear.

In this study, we used U-ExM to examine the location and function of the outer kinetochore proteins, *Pf*NDC80 and *Pf*Nuf2, and

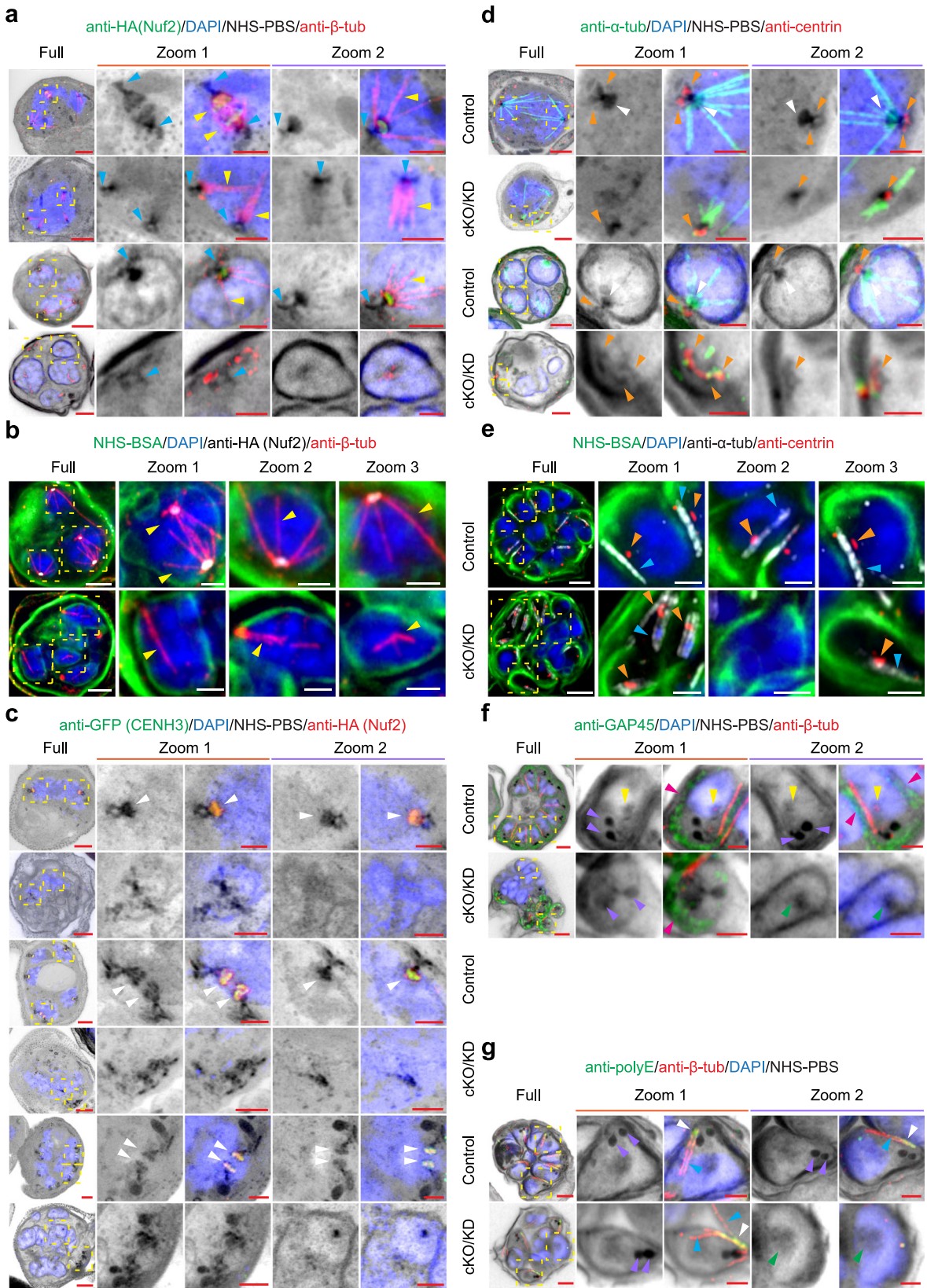

their interactions with other components of the mitotic machinery. The initiating step in mitosis is the formation of the hemispindle consisting of about five bundles of microtubules, emanating from a nuclear lumen-facing protein-dense structure (NHS-PBS-labelled in U-ExM)[17,20,23]. In our study, we see *Pf*NDC80 and *Pf*Nuf2 are concentrated at the base of the hemispindle. Interestingly the

centromere marker, *Pf*CENH3, is also concentrated in the same region indicating pre-assembly of the kinetochore/ centromere complex prior to chromatin duplication. This may facilitate the capture of the uncondensed chromosomes. This pre-assembly contrasts with mammalian kinetochores that assemble only during mitosis[42]. The co-incidence of *Pf*NDC80, *Pf*Nuf2 and *Pf*CENH3 with

**Fig. 5 | *Pf*Nuf2 disruption physically separates the two domains of the centriolar plaque, but microtubule polymerising capacity is retained. a, b** Representative U-ExM images of control and *Pf*Nuf2-cKO/KD (conditional knockout/knockdown) parasites labelled with the general protein dye NHS-PBS (**a** inverse greyscale) or NHS-BSA (**b** green), at early (30 hpi) and mid (36 hpi) schizont stages. The microtubules were labelled with anti-β-tubulin (yellow arrowheads, anti-β-tub, red), chromatin is marked by DAPI (blue), and the *Pf*Nuf2-HA is labelled with anti-HA (**a** green or **b** greyscale; evident only in control samples). Aqua arrowheads indicate the NHS-PBS-labelled outer centriolar plaque structures. The experiment was performed three times. **c** U-ExM images of *Pf*Nuf2-HA/GFP-*Pf*CENH3 control and cKO/KD in early (30 hpi), middle (36 hpi), and late (38 hpi) stage schizonts labelled with NHS-PBS (inverse greyscale). The kinetochore protein, *Pf*Nuf2, is labelled with anti-HA (red) and the centromere protein, *Pf*CENH3, is labelled with anti-GFP (green, GFP- *Pf*CENH3). The anti-GFP signal overlaps with the anti-HA signal (white arrowheads) in control cells. Both signals are lost in cKO/KD parasites. The experiments were performed three times. Early (30 hpi), mid (36 hpi) and segmented schizonts (41 hpi) were labelled with NHS-PBS (**d**) or NHS-BSA (**e**). The

outer centriolar plaque is labelled with anti-centrin (red, orange arrowheads). Both spindle microtubules (**d** green, white arrowheads) and subpellicular microtubules (**e** greyscale, aqua arrowheads) are labelled with anti-α-tubulin (anti-α-tub). The experiment was repeated three times. **f** Control and *Pf*Nuf2-disrupted segmented schizonts are labelled with NHS-PBS (inverse greyscale), highlighting the protein-dense rhoptries (purple arrowheads). The subpellicular microtubules are labelled with anti-β-tubulin (red), the chromatin with DAPI (blue) and the inner membrane complex with anti-*Pf*GAP45 (anti-GAP45, green, magenta arrowheads). An NHS-PBS-labelled region in control cells likely represents the spindle remnant/ inner centriolar plaque (yellow arrowheads). In *Pf*Nuf2-cKO/KO parasites, a green arrowhead indicates an unusual NHS-PBS-labelled area inside the nucleus. The experiment was performed two times. **g** Anti-polyE (green) is concentrated at the minus end of subpellicular microtubules (white arrowheads) close to the apically located rhoptries. The experiment was performed two times. All images are presented as z-projections. All scale bars are 5 μm, except the zoom images, which are 2 μm. Additional images of *Pf*Nuf2-cKO/KO parasites are presented in Supplementary Figs. 13, 14 and 15.

the NHS-PBS-labelled inner centriolar plaque, suggests that these kinetochore/ centromere components are likely major components of this structure. The lateral association of the kinetochores with the minus end of the hemispindle microtubules may stabilise microtubule bundles and help establish the spindle[44].

Following duplication of the outer centriolar plaque, a short mitotic spindle is established and *Pf*NDC80, *Pf*Nuf2 and *Pf*CENH3 relocate to the mid-plane of the spindle, consistent with the role of segregating sister chromatids[20,23,37]. The NHS-PBS-labelled region also moves along the spindle. The two outer centriolar plaque structures then migrate away from each other around the nuclear periphery, and an extended spindle is observed[37]. It is important to note that this is not the equivalent of the mammalian anaphase spindle as the chromatids have already been separated. At this stage, the kinetochore and centromere markers retract back to the region below the centrin puncta, and the NHS-PBS-labelled inner centriolar plaque structure is again evident. Taken together these data suggest that the NHS-PBS-labelled region in the nuclear lumen comprises the kinetochores and associated machinery. Dynamic movements of the kinetochores may help establish the spindle microtubules and help to achieve fidelity of DNA separation. The apparent plasticity of kinetochore organisation is consistent with previous live cell imaging studies in *P. berghei* and *P. falciparum*, as well as in other eukaryotic cells[17,20,23].

In agreement with previous reports[37,43,45], we confirmed that in later stages of schizogony, the outer centriolar plaque is positioned toward the apical complex (including rhoptries, IMC, PPM), which are in turn, position towards the periphery of the parent cell. Previous data for *P. falciparum*[37,43,45] and *T. gondii*[46,47] suggest that endoplasmic reticulum extensions close to the centriolar plaque generate the membrane material for the apical organelles. Based on U-ExM images, a physical link from the centriolar plaque to the apical prominence, the rhoptries, the Golgi and the apicoplast, has been suggested[37].

Previous electron microscopy studies examined fully formed merozoites and observed that the sub-pellicular microtubules arise from a tubulin-based apical annulus, called the apical polar ring[11,48,49]. Interestingly, our analysis of the earlier stage schizonts suggests that cytoplasmic microtubules are first initiated from the cytoplasm-facing part of the centriolar plaque, although we cannot rule out nucleation from a closely associated nascent apical polar ring. The suggestion that the outer centriolar plaque is the initial site of nucleation, is consistent with previous work on stage II gametocytes, which lack an apical polar ring, in which cytoplasmic microtubules appear to nucleate from the outer region of the nuclear membrane-embedded centriolar plaque[17]. It is possible that the formation of the cytoplasmic microtubules helps position and orient the nascent merozoite, similar to the role played by aster microtubules in vertebrate cells[50].

Conditional knockout/knockdown of *Pf*NDC80 and *Pf*Nuf2 reveals their important roles in *P. falciparum* mitosis. Consistent with recent *Pf*NDC80 mislocalisation data[29], we show that both proteins are essential for parasite proliferation in the asexual blood stage. Disruption of individual kinetochore components is expected to result in defective binding to the centromeres, and thus a failure to capture the chromosomes during the division process. In vertebrate cells, the presence of chromosomes that remain unattached to kinetochores is detected by the cell, activating signalling pathways that stall the onset of anaphase[51]. Interestingly, disruption of *Pf*NDC80 and *Pf*Nuf2 does not appear to prevent chromatin duplication or karyokinesis, although there is evidence for uneven inheritance of chromatin in the divided nuclei. Our data confirm that *Plasmodium* mitosis lacks robust checkpoints once mitosis has started, and that karyokinesis and cytokinesis are independently regulated processes[35,52,53].

In this study, we show that disruption of the kinetochore components leads to unregulated mitosis. In the early rounds of nuclear division, aberrant spindles are observed that are disorganised and disconnected from the nuclear membrane. Some of the newly formed nuclei exhibit multiple centrin puncta, while other nuclei have none, indicating uneven inheritance of the centriolar plaque, and dysregulation of nuclear division. Of particular interest, the signal for the centromere marker, *Pf*CENH3, is lost upon knockdown of kinetochore components. This suggests that kinetochore-based attachment of the chromosomes to the spindle microtubules is needed for integrity of the centromere complex.

During the final round of division, aberrant segmented merozoites are formed. The centrin-labelled outer centriolar plaque structure is disengaged from the cytoplasmic side of the nuclear membrane. Nonetheless, it remains connected to the apical organelles, such as the IMC and the rhoptries, and the sub-pellicular microtubules appear to be assembled from the disengaged centriolar plaque and the associated apical polar ring. The loss of this physical connection results in failure of the nascent merozoites to capture a nucleus. Intriguingly, the *Pf*NDC80 and *Pf*Nuf2 cKO/KD segmented schizonts are still capable of egress despite the incomplete assembly of the merozoites. The data further highlight the independent regulation of karyokinesis and cytokinesis in *P. falciparum* and provide functional evidence that the position of the outer centriolar plaque directs the location of apical complex assembly and merozoite segmentation.

The effect of NDC80 knockdown has been assessed previously in the gametocyte and mosquito stages of *P. yoelii*[30]. *Py*NDC80-knockdown did not affect the formation of spindle microtubules during exflagellation, of the *P. yoelii* male gametocyte but it did cause a defect in chromosome segregation, leading to a decreased number of activated male gametes[30]. The difference in phenotype may be due to stage- or species-specific effects.

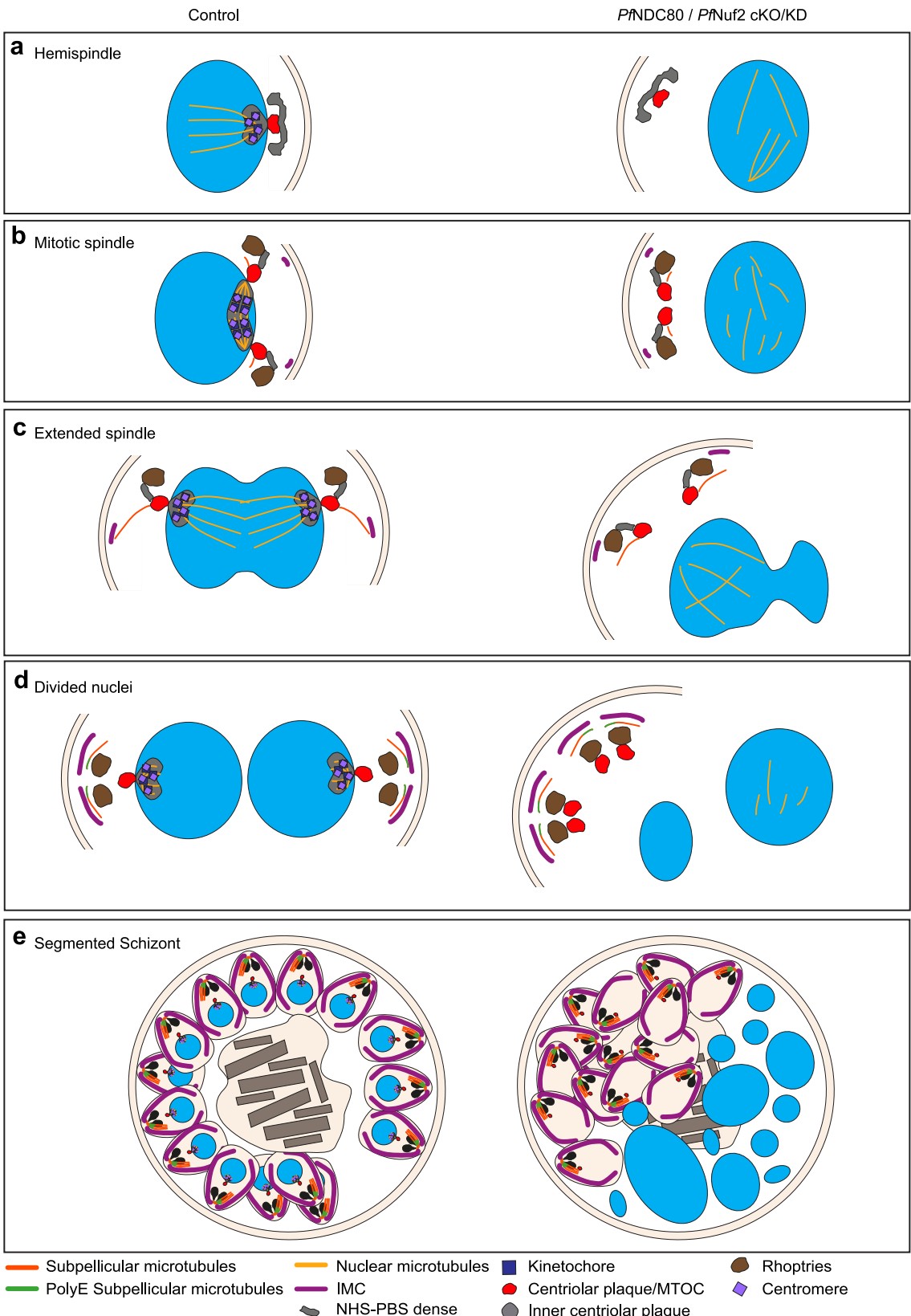

**Fig. 6 | Parasite spindle structures with and without kinetochore associated proteins.** Diagram illustrating the apparent locations of different parasite structures during mitosis; and disruption of organisation upon loss of the kinetochore components. (**a**) Hemispindle. (**b**) Mitotic spindle. (**c**) Extended spindle. (**d**) Divided nuclei. (**e**) Segmented schizont. In kinetochore-disrupted parasites, the mitotic spindle is disorganised, the inner and outer centriolar plaque structures are separated and the nexus between the mitotic machinery and the apical complex is lost.

The centriolar plaque mislocalisation phenotype that we observed is similar to, but more severe than, the phenotype observed upon conditional knock-down of Nuf2 in *T. gondii*[22]. *Tg*Nuf2 disruption decreased the interaction between the cytoplasmic part of the centrosome equivalent and the centrocone (spindle pole) during tachyzoite cell division. Moreover the daughter cell did not inherit chromatin and the centromeres remained assembled and associated with the parent nucleus[22]. Disruption of spindle and kinetochore-associated proteins (SKA), including *Tg*SKA1, *Tg*SKA2 and *Tg*SKA3, decreased the efficiency of the alignment and segregation of the centromeres and kinetochores into daughter cells[23], while knockdown of the SCF[FBXO1] E3 ubiquitin ligase also results in aberrant formation of the apical complex and the IMC[54]. Different phenotypes may be due to species-specific differences, such as the structurally different centriolar plaque in *Plasmodium* compared to the centrocone in *Toxoplasma*.

Less dramatic phenotypes have also been reported previously for disruptions of *P. falciparum* mini-chromosome maintenance complex binding-protein (MCMBP) and schizont egress antigen-1 (SEA1, a putative centromere-associated protein). *Pf*MCMBP-deficient parasites displayed hemispindle and spindle microtubules that were 30% longer, and showed more branches[35]. *Pf*SEA1- and *Pf*MCMBP-deficient schizonts exhibited 20% and 17.6% akaryotic merozoites, respectively[52,55]. By contrast, *Pf*NDC80-deficient mitotic schizonts exhibited 58% akaryotic merozoites.

In conclusion, this work provides insights into the role of the centriolar plaque as an important nexus between the intranuclear spindle microtubules and the apical complex - a connection that enables correct positioning of the mitotic spindle as well as positioning the apical complex and building the apical-basal polarity (diagram in Fig. 6). We show that deletion of the outer kinetochore components, *Pf*NDC80 and *Pf*Nuf2, results in disruption of the centromere marker and partial failure of the spindle microtubules to segregate the chromosomes. This, in turn, leads to disengagement of the outer centriolar plaque from the nuclear envelope and loss of the connection between the apical complex and the nucleus (diagram in Fig. 6). The breaking of this important nexus leads to aberrant mitosis, resulting in akaryotic merozoites that are incapable of invading new RBCs. Our findings are reminiscent of a study in human cells, in which disruption of the centromere complex was shown to impact mitotic spindle pole integrity, leading to release of the microtubule minus-ends from the centrosome[27]. Our findings underscore the evolutionary conservation of pathways involving centromeres and centrosomes and highlight the value of using *P. falciparum* as a model for exploring fundamental questions in cell biology. Targeting the kinetochore/ centrosome nexus may provide a strategy to fight this very important human disease.

## Methods

### Parasite culture and synchronisation
Asexual blood stage *P. falciparum* parasites were cultured in RPMI-HEPES containing 0.25% AlbuMAXII and 5% human serum at 37 °C[56]. The parental parasite line NF54 DiCre used for transfection was maintained in 2.5 ug/mL Blasticidin S[31]. Late-stage parasites were enriched from culture using Percoll purification (GE Health) or magnetic purification (Miltenyi Biotec). Sorbitol synchronisation was used to obtain ring stage parasites[57].

### Plasmid constructs and transfection
Transgenic parasite lines were generated in which *Pf*NDC80 and *Pf*Nuf2 were C-terminally tagged with a 3×HA tag and incorporated flanking *loxP* sites and a *glmS* riboswitch in the 3′UTR of the gene allowing for conditional knockout (cKO) of the gene and knockdown (cKD) of the transcript. The repair constructs were created by cloning a synthetic DNA fragment (HR1-intron-recodonised coding sequence,

Genewiz) consisting of a 5′ homology region, an artificial intron containing a *loxP* site and the recodonised 3′ end of the gene into the *Bgl*II/ *Pst*I sites of the pGLMS-HA-*glmS* plasmid[58]. The 5′ homology region of *Pf*NDC80 comprises 198 bp upstream of the start site and the first 171 bp of the coding sequence. For *Pf*Nuf2, 322 bp of the 5′ untranslated region and 36 bp of the coding sequence were used. These positions were chosen based on suitable Cas9 guide locations. All sequences downstream of the guide were recognised in the synthesised fragments. A 3′ homologous targeting sequence from the 3′ end (HR2) of *Pf*NDC80 or *Pf*Nuf2 was PCR amplified (Supplementary Table 1) and cloned into the *Eco*RI and *Kas*I sites of the plasmid. Plasmid DNA (50 μg) was linearised by overnight digestion with *Bgl*I and *Bgl*II enzymes, ethanol precipitated and resuspended in 30 μL TE buffer ready for transfection. Custom guides were designed and ordered as an Alt-R CRISPR-Cas9 crRNA (IDT™, Supplementary Table 2). Transfections were performed using the complexed Cas9 recombinant protein, guide RNAs and the linearised repair template[59].

A previously generated NF54 parasite line stably expressing a rapamycin inducible split Cre recombinase[31] was used as the parental strain in this work. These transfectants were maintained with 2.5 nM WR and 2.5 μg/mL Blasticidin S. Integration of the plasmids into the endogenous loci and *loxP* excision were confirmed by PCR (Supplementary Figs. 1b, c and 4c, e) using the primers described in Supplementary Table 1. See Supplementary Fig. 1a for schematic and primer locations. Uncropped gels are provided in Supplementary Fig. 16a, b, f, h.

To investigate the location of *Pf*CENH3, we co-transfected the *Pf*Nuf2 and *Pf*NDC80-HA transfectants with a plasmid encoding a GFP-*Pf*CENH3 chimeric protein. The *ydodh* gene was cut from the pNLS-FRB-mCherry plasmid[60] using the BamHI and HindIII restriction sites and directionally cloned into the BamHI and HindIII sites of the p*Pf*NDC80-GFP-*Pf*CENH3 construct[17], replacing the selectable markers. The GFP-*Pf*CENH3 plasmid was transfected into the *Pf*NDC80 and *Pf*Nuf2 parental strain and maintained on 0.9 μM DHODH inhibitor (DSM1)[61].

### *P. falciparum* growth assays
Tightly synchronised (2-h window) ring-stage parasites were plated in triplicate into 24-well plates (2 mL/well) at 0.2–0.5% parasitemia (5% haematocrit). Parasite-infected RBCs were treated with either 100 nM rapamycin (41-h duration) and 2.5 mM glucosamine (continuously) or with 0.04% DMSO (41-h duration). Parasite samples were taken at 36 h from the first cycle and then every 41 h for 4 cycles. Parasitemia levels were assessed by flow cytometry. Parasite-infected RBCs were diluted to 0.01% haematocrit in 1×PBS and labelled with SYTO™ 61 for 15 min and analysed by flow cytometry[62]. Flow cytometry was performed on a BD FACSCanto™ II equipped with a High Throughput Sampler (HTS) and data was analysed using FlowJo software (version 10.7.1_CL)[62].

### Western blotting
Tightly synchronised schizonts were harvested at relevant time points and lysed with chilled saponin (0.03% (w/v) in PBS buffer) and incubated on ice for 15 min. The lysate was centrifuged at 3300 × *g* for 10 min, and the supernatant was removed and discarded. The parasite pellets were resuspended and washed three times in pre-chilled PBS containing an EDTA-free protease inhibitor cocktail (cOmplete, Roche). The samples were prepared for electrophoresis by resuspending the parasite pellets (10 μL) in Bolt LDS sample buffer (4×) (Invitrogen) (25 μL), Bolt sample reducing agent (10×) (Invitrogen) (10 μL) and chilled PBS containing protease buffer (65 μL). The samples were heated at 95 °C for 10 min and loaded into 4–20% Mini-PROTEAN® TGX™ Precast protein gel (BIO-RAD) and separated by electrophoresis in 1× Tris/Glycine/SDS buffer (BIO-RAD). Proteins were transferred from gels onto nitrocellulose membranes using the Trans-Blot® Turbo™ transfer system. The membranes were blocked in

Intercept® (PBS) Blocking Buffer (LICORbio) for an hour at room temperature. Both primary and secondary antibodies were diluted in the blocking buffer. Primary antibodies: mouse anti-HA (Millipore; 05-904; LOT: 3005257; 1:500), mouse anti-GFP (TakaRa Living Colors® Av Monoclonal Antibody, JL-8 (632381); 1:500) and rabbit anti-*Pf*aldolase (abcam, ab207494; 1:3,000)[63]. Secondary antibodies: goat anti-mouse IRDye® 800CW (LICORbio; 926-32210; 1:10,000), donkey anti-rabbit IRDye® 680LT (LICORbio; 925-68023; 1:10,000). The membranes were incubated in primary antibodies overnight at 4 °C and secondary antibodies for 1 h at room temperature. The membranes were washed three times with phosphate buffered saline with 0.05% TWEEN® 20 (PBST) after incubations with primary and secondary antibodies, respectively. The membranes were imaged on an Odyssey® CLx imaging system (LICROBio). Uncropped gels are provided in Supplementary Fig. 16c, d, e, g, i.

### Fluorescence microscopy

For the live cell imaging, tightly synchronised mature schizonts (41 hpi) were collected. The parasites were incubated with Hoechst 33342 (5 µg/mL, Invitrogen) to label DNA, and Tubulin Tracker Deep Red (1 µM, Invitrogen) to label the microtubules, for 15 min at 37 °C, before imaging. The live cell data were acquired on a restorative wide field deconvolution microscope (DeltaVision DV Elite, Applied Precision) equipped with a 100× oil immersion objective (numerical aperture (NA) = 1.46). The images were acquired and deconvolved using the softWoRx 6.1 software (Applied Precision). The images were processed in FIJI software (version 2.3.0). All live cell images are displayed as maximum intensity projections (using FIJI) of full z-stacks.

For immunofluorescence microscopy, highly synchronised mitotic (36 hpi) schizonts were sampled onto coverslips pre-coated with 0.1 mg/mL erythroagglutinating phytohemagglutinin (PHAE) and incubated at 37 °C for 20 min. The cover slips were washed three times with PBS, prior to being fixed in 2% (v/v) formaldehyde in PBS for 30 min at 37 °C. After fixation, the coverslips were washed three times in PBS and bound cells were permeabilized with 0.2% (v/v) Triton-X 100 (in PBS) for 20 min at room temperature. The coverslips were again washed three times with PBS and incubated with the primary antibodies (diluted in 3% BSA/PBS) for 3 h at room temperature. The following primary antibodies were used: rabbit anti-HA (Sigma-Aldrich; H6908; 1:200) and mouse anti-β-tubulin (Sigma-Aldrich; T5201; clone TUB 2.1; 1:200). After washing three times with PBS, the coverslips were incubated for 1 h with secondary antibodies (diluted in 3% BSA/PBS). The following secondary antibodies were used: Alexa Fluor 568 goat anti-mouse IgG (H + L) (Invitrogen; A-11004; 1:500) and Alexa Fluor 488 goat anti-rabbit IgG (H + L) (Life technologies; A-11008; 1:500). The parasite DNA was labelled with DAPI solution (2 µg/mL in PBS, Thermo Scientific, 62248) for 15 min at room temperature and washed three times with PBS. The coverslips were mounted in anti-fade solution (90% glycerol/0.2% w/v p-phenylenediamine) on glass slides. The images were acquired and processed as described above. All immunofluorescence images are displayed as maximum intensity projections (using FIJI software) of full z-stacks.

### Ultrastructure expansion microscopy

U-ExM was performed following published protocols[33–36] with some adaptions as follows. Coverslips (22 × 22 mm # 1.5 mm; Lecia, 3800106) were cut into four square pieces (11 × 11 mm). The coverslips were incubated with poly-D-lysine (Gibco, A38904-01, 0.1 mg/mL) for 1 h at 37 °C, washed three times with MilliQ water, dried under nitrogen gas and placed in a 24-well plate. Tightly synchronised mitotic (30, 34, 36, 38 hpi) or mature segmented schizonts (41 hpi; with a 2-h window) were harvested by Percoll purification, resuspended in PBS and overlayed onto the precoated coverslip and incubated for 15 min at 37 °C. The unbounded cells were carefully removed from the well and the bound cells fixed with 4% (v/v) formaldehyde in PBS for 15 min at 37 °C.

After fixation, the coverslips were washed three times with prewarmed PBS and incubated in 1.4% (v/v) formaldehyde/2% (v/v) acrylamide in PBS overnight at 37 °C.

A monomer solution comprising sodium acrylate (19% w/w; Sigma-Aldrich, 408220), acrylamide (10% v/v; Sigma-Aldrich, A4058) and N, N'-methylenebisacrylamide (BIS; 2% v/v; Sigma-Aldrich, M1533) in PBS was prepared 24 h before use and stored at −20 °C. For parasite embedding in the gel, 5 µL of pre-chilled 10% (v/v) tetra-methylenediamine (TEMED) (Thermo Scientific, 17919) and 5 µL of 10% (w/v) ammonium persulfate (APS) (VWR Chemicals BDH, BDH9216) were added to 90 µL of monomer solution and vortexed for 3–5 s. Parafilm squares larger than the coverslip were placed on ice and are used as a base for gel setting and sample embedding. Thirty-five microliters of the gel mixture were added onto the prechilled Parafilm and the coverslip (cell side face down) placed on top of the gel mixture and incubated at 37 °C for 1 h. The gel and coverslip were removed from the mould and incubated in denaturation buffer (200 mM sodium dodecyl sulphate (SDS), 200 mM NaCl, 50 mM Tris, pH = 9) in a 6-well plate and incubated, shaking for 15 min at room temperature. The gels were detached from the coverslips and transferred into 1.5 mL Eppendorf tubes containing the denaturation buffer and incubated at 95 °C for 90 min. The denatured gels were transferred into a clean dish (10 cm) containing MilliQ water for the first round of expansion. The dishes were placed on the shaker for 30 min at room temperature, the MilliQ water exchanged, and the dishes again placed on the shaker. This step was performed three times in total. After expansion, the gels were re-equilibrated by incubating with PBS for 15 min and then blocked in 3% BSA/PBS for 30 min at room temperature. After blocking, the gels were incubated with primary antibodies at room temperature overnight on the shaker (Supplementary Table 3). Following antibody incubation, the gels were washed three times with PBST (10 min/wash), prior to addition of the secondary antibodies (Supplementary Table 3), general protein dye DyLight™ 488 NHS Ester (Thermo Fisher, 46402) and DAPI for 2.5 h incubation in the dark with gentle shaking at room temperature. For experiments where membrane staining was required, the secondary antibodies, DAPI (200 µg/mL final concentration) and NHS Ester dye (10 µg/mL final concentration) were diluted in 3% BSA/PBS solution, while for more generalised protein labelling, the secondary antibodies, DAPI (200 µg/mL final concentration) and NHS Ester dye (10 µg/mL final concentration) were diluted in the PBS. Following secondary antibody incubation, the gels were washed three times with PBST (10 min/wash) and transferred to a Petri dish containing MilliQ water and incubated 30 min. The MilliQ water was exchanged three times.

To mount and image the expanded gels, 35 mm coverslip dishes (uncoated, γ-Irradiated, MarTek Corporation, P35G-2-14-C-GRID, or uncoated, #1.5 polymer coverslip, hydrophobic, sterilised ibidi, 81151) were coated with 0.5 mL poly-D-lysine for 1 h at 37 °C, washed with MilliQ water and dried under nitrogen gas. The gels were transferred to precoated glass bottom dishes, with the cell side down. The expansion of each gel was calculated by measuring the length and width of gel prior to and after expansion.

U-ExM data were acquired using a Zeiss Elyra LSM880/ Axio Observer Z1 LSM980 microscope equipped with an Airyscan detector (Carl Zeiss) in "SR" mode and using a 100 × oil immersion objective (numerical aperture = 1.4) or 63 × oil immersion objective (numerical aperture = 1.4) or a Zeiss Axio Observer LSM900 with an Airyscan 2 detector with a 63x APOCHROMAT objective (numerical aperture = 1.4). The images were acquired as Z-stacks (0.2 µm or 0.3 µm per slice) and processed using the Airyscan Processing tool within the Zen black software (version 2.3) or Zen blue software (version 3.8). All U-ExM images are displayed as maximum intensity projections (using FIJI software) of multiple z-slices (thickness ranges from 2 µm to 3 µm). Nuclear volumes (expanded) were analysed in segmented schizonts, using the Imaris software (version 10.0.0). Airyscan processed images

were converted to Imaris file format (.ims) using Imaris File Converter (version 10.0.1). Prior to 3D reconstruction of the nuclei, the membrane channel (NHS-BSA staining) was subtracted from the nucleus channel (DAPI staining) using the Channel Arithmetics tool to improve the boundary detection between the nuclei. The remaining nucleus signal (*i.e.*, DAPI minus NHS-BSA) was used to reconstruct a 3D object using the 'surface creation wizard'. In all images, an automatic threshold was used to avoid user bias in volume measurement and the reconstructed 3D objects were inspected manually to avoid over/ under detection in counting.

## Statistical analysis

All data are presented as means plus and minus the standard deviation (SD), with specific details outlined in the Fig. legends. Each live-cell fluorescence experiment and flow cytometry growth assay were performed at least three times. Each PCR and Western blot were performed at least two times. The data displayed are representative of these experiments. Not all expansion microscopy experiments were repeated three times, due to the large volumes of antibodies required and availability. This is recorded in the Fig. legends. The cell numbers analysed in each experiment are presented in the Fig. legends. The analysis of the volumes of the (expanded) nuclei is described in the Fig. legend. Statistical significance was analysed using Welch's t-test with 95% confidence intervals as indicated in the Fig. legends. Statistical analyses were conducted using GraphPad Prism 8.

## Transmission electron microscopy

Segmented schizonts (41 hpi) were enriched by Percoll purification. The enriched parasite-infected RBCs were fixed in 1 mL of 2% (v/v) glutaraldehyde and 2% (v/v) formaldehyde in 1 × PHEM buffer (60 mM PIPES, 25 mM HEPES, 10 mM EGTA, 2 mM $MgCl_2$) and incubated overnight at 4 °C. The cells were washed for 5 min in 1.5 × PHEM buffer, centrifuged at 500 × g, the buffer was replaced, and the infected RBCs were washed a further 2 times. The parasites were post-fixed in 1% (w/v) osmium tetroxide in 1 × PHEM buffer for 30 min in the dark. The infected RBC pellet was washed with 1.5 × PHEM buffer three times as described above. The infected RBC pellet was next incubated in 1% (w/v) tannic acid in 1 × PHEM buffer for 20 min. Samples were washed three times with MilliQ water, with centrifugation as described above. The infected RBCs were next incubated in the dark in 1% (w/v) aqueous uranyl acetate for 20 min, before 3 rounds of washing in MilliQ. Following washing, samples were dehydrated in an ascending ethanol series (30%, 50%, 70%, 90%, 100%, 100%) (5 min each condition). Samples were next incubated in two changes of absolute acetone for 10 min each. The cells were gradually infiltrated with acetone: medium grade Procure 812 resin (2:1, 1:2) for 2 h, followed by pure resin incubation overnight and a final change of the pure resin the next day for another 3 h, prior to polymerisation for 48 h at 60 °C.

After polymerisation, samples were trimmed and 70 nm sections generated using an ultramicrotome (Lecia EM UC7; Leica Microsystems, Wetzlar, Germany). The sections were collected on formavar-carbon 100-mesh and imaged on a FEI Tecnai F30 electron microscope (FEI company, Hillsboro, OR) at an accelerating voltage of 200 keV.

## Array tomography

For array tomography, 120 nm serial sections were generated and collected on a hydrophilized silicon wafer. Samples were post-stained with 2% aqueous uranyl acetate for 10 min and Reynold's lead citrate for 10 min. Backscattered field emission scanning electron microscopy was performed using a Zeiss Sigma operating at 3 keV accelerating voltage and 5 mm working distance. Images were normalised and binned in FIJI and automatically aligned using the StackReg plugin[64]. Features of interest were segmented, visualised and analysed using IMOD (version 4.9.9)[65].

## Reporting summary

Further information on research design is available in the Nature Portfolio Reporting Summary linked to this article.

## Data availability

Additional data are available in Supplementary Information. Source data are provided. Source data are provided with this paper.

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

## Acknowledgements

We thank Professor Rita Tewari, the University of Nottingham, UK and Professor Jing Yuan, Xiamen University, China, for advice. We acknowledge the facilities at the Biological Optical Microscopy Platform and the Flow Cytometry Platform, The University of Melbourne, the Microscopy Resources on the North Quad (MicRoN) core at Harvard Medical School, and the University of Sydney Microscopy & Microanalysis facilities of Microscopy Australia. For research support, we thank the Australian Research Council (FL150100106 to L.T.), the National Health and Medical Research Council of Australia (APP1098992 to L.T.) and the NIH (5R01AI167570 to M.T.D). B.L. is supported by an American Heart Association Postdoctoral Fellowship (23POST1011626).

## Author contributions

Conceptualisation: J.L., G.J.S., B.L., S.A., M.W.A.D., and L.T.; Investigation: J.L., G.J.S., and E.C.; Analysis: J.L., G.J.S., F.B., E.C., M.W.A.D., and L.T.; Funding acquisition: F.B. and L.T.; Writing: J.L., G.J.S., F.B., E.C., B.L., M.T.D., S.A., M.W.A.D., and L.T.

## Competing interests

The authors declare no competing interests.
