## [Peer Review File · Nature Communications]

Disruption of Plasmodium falciparum kinetochore proteins destabilises the nexus between the centrosome equivalent and the mitotic apparatusREVIEWER COMMENTS

Reviewer #1 (Remarks to the Author):

In this article the authors investigate the localisation and requirement of two kinetochore proteins, Nuf2 and Ndc80, during the asexual replication of *Plasmodium falciparum* inside erythrocytes. They first describe the expected dynamic of kinetochore proteins along the mitotic spindle at various stages during schizogony. They then describe the effect of depletion of kinetochore proteins on the stability of the centriolar plaque. They further discuss this dynamic in relation with the apical complex that is linked to the mitotic machinery with a bipartite MTOC. They finally investigate the effect of Nuf2 or NDC80 deletion/depletion during schizogony. As expected this strongly affects DNA partitioning and the stability of the mitotic spindle, as observed in most eukaryotic systems. Most surprisingly they describe the disruption of the outer and inner centriolar plaques that in the last round of replication leads to aberrant cytokinesis.

The images shown in this article are beautiful and give a nice cellular overview of both the localisation of kinetochore proteins and their cellular requirement. However, the authors focus too much on the relationship between kinetochores and the apical complex blurring the main finding of this work. The link between the mitotic machinery and the apical complex in apicomplexans has been described, see for example PMID11085247 (2000) or PMID17604449 (2010) for a review. While I understand that implementation of expansion microscopy with NHS ester staining represents an accessible and powerful approach to visualise these structures, a similar description of the link between the mitotic machinery and the apical complex has already been performed comprehensively and well-reviewed by some of the co-authors (PMID 36993606 and 37571814) as was the relative position of kinetochore proteins with the centriolar plaque (PMID36006241).

Major general comments

While I understand that this represents an exciting and hot area of research, I think that by focusing on the link between the kinetochore and the apical complex the authors are avoiding to tackle the most relevant phenotype described in this article, ie, the destabilisation of the "nexus" between the apical complex and the mitotic spindle: the intervening MTOC. Non-specialist readers will likely be misled towards a functional link between the kinetochore and the apical complex, which is very indirect. In that respect the title is particularly misleading by highlighting the link between the "nexus", aka centriolar plaque, and the apical complex, which is already known.

At a technical level – expansion microscopy has been instrumental to describe the effect of knockdown of kinetochore proteins in these cells. When coupled with immunolabelling of specific proteins, this technique opens the doorway to a medium throughput method at sub-EM resolution to phenotype mutants in a way that is not possible by EM. Instead the authors focus on NHS ester and tubulin staining throughout the work, meaning the structures described would actually be seen at greater resolution by EM, and this would likely avoid confusion as to where each "continuous structure" delineates. Therefore, I am uncertain as to advantages here of using UExM compared to EM for their phenotyping. One example of this is the "dark spots" in the nucleus (as seen by NHS ester alone following UExM). How do you know these are not clusters of kinetochores? An anti-CENH3 may answer this – either stain these spots appropriately to gain the advantages of UExM or just use EM for greater resolution and indication of what they are composed of.

In that respect, I think the authors should recenter the wording of the paper on the stability of the centriolar plaque relative to the kinetochores and consolidate the phenotyping of the rupture of the centriolar plaque.

More specific comments

line 40 - change "substantive" to "substantial"

line 47 - change "divergent" to "different"

line 51 - what is "semi-synchronised"?

Line 72 – Please also reference PMID36006241.

Line 106 and supplementary Figure 1d legend - how was the 90% reduction of protein levels quantified to support "very efficient and sustained knockdown"? It is also not clear to what this is compared? For example, in *Toxoplasma gondii*, auxin-induced depletion often yields depletion to undetectable levels by western blotting and this is directly compared in discussion line 422. If this technology is to be compared, the data must support this claim. Please show a dilution series to support this 90% value. In addition, it is not indicated on which stages the blots are performed. Without timing depletions, it is not possible to interpret which mitosis is affected during schizogony or the possibility of any delays.

Supplementary Figure 1a - primer binding sites and expected sizes are missing from schematic

Supplementary Figure 1b - expected sizes are missing from gels

Supplementary Figure 1c - time post induction is missing from western blot. How long does it take to achieve this level of reduction?

Supplementary Figure 1f legend - what is "conventional immunofluorescence microscopy"

Line 183 - "There is evidence for a connection from the rhoptries through to the NDC80-labelled inner centriolar plaque, as evidenced by a continuous strong NHS". I find this statement misleading. There are many structures between the rhoptries and kinetochores. These do not form single continuous structure. EM has previously clearly shown these structures, and also does in figure 4F.

Lines 185 - "The NHS-PBS-labelled structure extends towards the PPM and appears to seed the formation of the inner membrane complex (IMC), apical complex and the cytoplasmic microtubules, in a process that appears to be driven by the positioning of the centriolar plaque (Fig. 2d)." Similarly, whilst Fig 2d shows a very nice image of the spindle and kinetochores (both NHS ester and immunostained), I do not believe this claim is supported. The overall structure is linked by multiple components, also including the "centriolar plaque" (a term coined by EM and not NHS ester staining).

Line 199 – the IMC and the PP are not part of the apical complex, they form the pellicle.

Line 223 - "We investigated if PfNDC80 cKO/KD affects the final round of nuclear division (38-40 hpi) when the cell initiates cytokinesis and the nuclei are packaged into nascent merozoites". See earlier point regarding levels and timing. Without evidence of how long it takes for these proteins to be depleted by the cKO/KD method, it is not possible to know which round of nuclear division is affected. What if 50% protein levels are sufficient for mitosis in these cells and below 50% is only achieved after 35 hours?

Line 231 and line 256 - The above comment would also support the partial KD fitness conferring phenotypes observed.

Line 296 "Inspection of the NHS-PBS-labelled features in PfNuf2-cKO/KD schizonts, reveals disruption of the centriolar plaque structure with a loss of the physical connection between the intra- and outer-nuclear structures". While this is a central statement of the paper I do not see this by these images and do not believe this can be shown by NHS ester staining as long as no specific markers of the outer and inner centriolar plaque are shown. Please show EM equivalents to support this claim.

Line 387 - "Interestingly, disruption of PfNDC80 and PfNuf2 does not appear to halt chromatin

duplication, karyokinesis or stall mitosis". Some of the most severe SAC mutants in human cells e.g. (PMID21664570) only delay entry into anaphase and do not provide a complete block. No experiment described in this paper have looked at the length of mitosis in these cells.

Line 398 - same as above, there is no clear evidence of the inner centriolar structure by NHS ester staining.

Line 406 – The FBXO1 protein was previously shown to localise on between the forming rhoptries and the outer centriolar plaque and its depletion also led to aberrant formation of the apical complex and the IMC (PMID31348812 36898988).

Reviewer #2 (Remarks to the Author):

During the intraerythrocytic asexual stage, the Plasmodium parasites proliferation in a way called schizogony, in which several asynchronous rounds of endomitosis are followed by one round of karyokinesis and cytokinesis, forming a segmented schizont containing several daughter haploid nucleated merozoites. The mechanism underlying the precise separation and allocation of replicated chromosomes into daughter nucleus, the precise structure linking the nucleus and apical organelles are yet to be verified. The authors used ultrastructure-expansion microscopy and high-resolution imaging to explore the connection between nuclear spindle microtubules and cytoplasmic sub-pellicular microtubules during the asexual proliferation of Plasmodium falciparum. They found that the centriolar plaque, the chromosome-capturing structure kinetochore and apical complex are associated during the early stage of mitosis, functions as a nexus coordinating the formation of nuclear spindle and sub-pellicular microtubules. This is the first study in the Plasmodium to reveal the relationship between the nuclear spindle and the cytoplasmic sub-pellicular microtubules in the asexual stage. Furthermore, the authors investigated the role of two kinetochore proteins NDC80 and Nuf2 during schizogony in Plasmodium falciparum.

However, I have some comments and suggestions for the authors of this manuscript prior to its publication.

Major comments :

Line 59: "Aster fibres form on the other side of the centrosomes and radiate towards the poles of the dividing cell to position the spindle apparatus during mitosis." It is not described clearly to the reader.

Line 199-200: The conclusion is overstated and not supported by the experiment evidences. The NHS ester-conjugated, sulfonated fluorophores are water-soluble probes that unspecifically reacts with primary amines in proteins and other molecules to yield stable amide bonds. The imaging via this generalized dye can only tell that these structures (with dense signal) are associated, it is overstated that these structures are physically connected.

In the Figure 1d, the staining signal of the beta-tubulin (for the mitotic spindle) is not clear. In the current image, no clear microtubules (radiating form) could be seen. It is hard to judge the relative localization between Nuf2 protein and spindle microtubules. Suggest to replace with a better image.

In the Figure 1c and d, the kinetochore proteins NDC80 and Nuf2 were detected to be localized at the base site of spindle. No protein signal was seen along the spindle microtubule or at the plus end of microtubule. Based on the kinetochore localization compared to spindle, it is hard to speculate the roles of kinetochore in the separation of duplicated chromosomes. It is good to discuss this in the discussion section.

Fig 2b: PBS-NHS-labelled structure (Centrin is associated with a punctate NHS-PBS-labelled structure just outside the nucleus) is not clear.

If the kinetochore is not functional, how the chromosomes could be separated after duplication? In

the Figure 3, 4, and 5, both the NDC80 cKO/KD and Nuf2 cKO/KD parasites displayed defected spindle microtubules inside the nucleus, decoupled connection between spindle and sub-pellicular microtubules, and daughter merozoite without nucleus. However, in the schizonts of the mutant parasites (NDC80 cKO/KD and Nuf2 cKO/KD) shown in these figures, most of the nucleus were successfully separated after chromosome duplication. Have the authors observe any schizont containing a big duplicated nucleus without no separation.

Minor comments :

Line 62: The authors mention that "By contrast, *P. falciparum* does not have centrioles, nor any other distinct structure within the centrosome equivalent, which is termed the centriolar plaque." However, throughout the manuscript, inner centriolar plaque and out centriolar plaque were shown several times. It is better to say "no canonical centrioles in Plasmodium"

FigS1d-e: For the 90% decrease in the expression of both NDC80 protein and Nuf2 protein described in the figure legend, a quantitation analysis should be added in the figure.

FigS1d-e: As ERC is an internal reference protein, the description of this internal reference protein needs to be added.

Line 120: A citation should be provided for "the protein-rich structures, such as the rhoptries".

Line 275: "•••positioned at one end of the merozoite and the nucleus occupying the basal region". Rephrase this sentence for good understanding.

Reviewer #3 (Remarks to the Author):

This study provides enticing insight into the positioning and function of the outer kinetochore proteins of the malaria-causing parasite *Plasmodium falciparum*. While the very first sections of the manuscript reiterate some previously describe organization principles of the mitotic machinery the authors uncover a highly intriguing phenotype resulting from kinetochore protein depletion. They note a loss of centriolar plaque integrity resulting in spindle misorganization and detachment of the nuclei from their plasma membrane-localized budding sites. While the presented findings are specific to the apicomplexan field this work highlights the divergence of eukaryotic cell division biology and how canonical components are differentially used by malaria parasites to drive their atypical nuclear multiplication and segregation. The authors make good use of the now well-established ultrastructure expansion microscopy protocol while introducing a small variation that strongly accentuates the visibility of membranes, including the nuclear membrane. They also implement less commonly used array tomography to reveal more phenotypical details in their mutants. The presented study is coherent, and the conclusions and claims made in the manuscript are supported by the data. Yet some additional analysis might be required to clarify the mechanistic pictures the authors are drawing about the roles of kinetochores in centriolar plaque integrity, which I will outline below.

Major comments

1) In the figures 1 and 2 the authors provide new details about the positioning of kinetochores within dividing nuclei of asexual blood stage parasites. They also reiterate several findings that have been made previously concerning kinetochore dynamics (Brusini et al.), inner and outer centriolar plaque structure (Liffner et al. and Simon et al.), and microtubule organization (Balestra et al.). Yet they repeatedly use a wording implicitly suggesting that their study is the first to show this as in line 132ff, but more specifically line 199ff with statements like "...these data indicate a physical connection from the spindle and inner centriolar plaque..." and "data further suggest that the spindle microtubules nucleate from a protein-dense structure...". Both findings having been described previously. Similarly in the discussion (lines 336 ff) the authors use the wording "We showed..." going on to describe previously published data. Even though the authors conclude the respective result paragraphs (lines 142 & 205) with the mention that their data is consistent with

previous publication this does not clarify sufficiently to a non-expert reader the degree of novelty of their contribution. I understand that it is important to reiterate those findings for comprehensiveness, but strongly suggest rewording of the aforementioned sections accordingly using words like confirm / validate.

2) A potential discrepancy between the data shown in this manuscript (Fig. 1 & 2) and previous findings is the positioning of the kinetochore signal in the hemispindle stage. The authors show a positioning within the inner centriolar plaque, which has been previously described as chromatin-free. Further CenH3, the centromeric histone onto which kinetochore components bind, has been shown to localize more on the edge of the inner centriolar plaque (Simon et al.). One way to resolve this discrepancy is that potentially some of Ndc80 is not chromatin bound or maybe that the kinetochore as a whole significantly extends into the protein dense region. A key difference to previous publication is that this study focusses more on later stages of nuclear division. The authors should comment on this, but more importantly show a few more images of control cells and mutants of early schizont stages with 1-3 nuclei i.e. during their first nuclear divisions. This is relevant a) to resolve the aforementioned discrepancy, b) since kinetochores are likely already required for the first nuclear division it is critical to analyze a potential early phenotype to clarify whether a later phenotype is a just an "escalation" of previous failures to e.g. segregate chromosomes c) Early and late schizont stages are differently organized, i.e. by presence/absence of rhoptries and subpellicular microtubules (Liffner et al), nuclear size and positioning, and should therefore be investigated separately.

3) An aspect of this intriguing study that needs clarification is whether the parasites experience a chromosome segregation defect due to the well described function of outer kinetochore proteins in this context. Visually Fig. 3c suggest that there might indeed be some errors in partitioning DNA, which is coherent with the increased nuclear volume observed in a subset of nuclei (Fig. S8). Yet is unclear to me how this can be reconciled with the equal final amount of nuclei observed while the number of formed merozoites (including empty merozoites) is lower in the mutant. I would not go as far as suggesting FISH analysis to quantify chromosome segregation errors, but maybe the authors can attempt a more thorough quantification of absolute DNA content in postmitotic nuclei (preferably using early schizont stages) to address this.

4) In Fig. 5a the authors provide, an interesting NHS-PBS staining for the Nuf2 knock down. It shows a protein dense region, supposedly the inner CP, which is detached from the nuclear periphery while still being associated with polymerizing microtubules. This is very intriguing and a critical piece of evidence that the kinetochore "nexus" is essential for CP integrity. Therefore, the authors should provide the same analysis for the Ndc80 knock down. Consequently, I feel there is a discrepancy between model Fig. 6 and Fig. 5a where NHS-PBS staining suggest that the detached inner CP is still the site of microtubule nucleation, while in the model these are not drawn as overlapping. Please clarify the model accordingly once evidence for the Ndc80 KD has been added.

Minor comments

5) Line 134: Please change the term "interpolar spindles" to "anaphase spindle" (or "extended spindle"). The term interpolar refers to a subset of microtubules within a spindle, which fully connect the two poles (hence interpolar), not to the spindle as a whole (which contains several "categories" of microtubules).

6) Please briefly quantify the spindle microtubule phenotype observed in Fig. 3b. A mention of how many of the total imaged nuclei/cells display this phenotype in the text would suffice.

Response to Reviewers.

We appreciate the comments of the reviewers.

As requested by the reviewers, we have provided additional data in support of our findings. We have generated two additional transfectant parasite lines that now provide a marker for the centromere (GFP-*PfcenH3*), expressed in the *PfNDC80*-HA and *PfNuf2*-HA cKD/KO backgrounds. This enables us to provide the complementary experimental support requested by reviewers #1.

We have undertaken studies of early schizonts as requested by reviewers #1, as well as provided additional data for NHS-PBS-labelled *PfNuf2* cKD/KO, as suggested by reviewer #3.

We have also provided additional characterisation of the time dependant excision and knockdown of the conditional mutant lines, as suggested by reviewer #1.

We thank the reviewers for their suggestions as these additional experiments pointed to an unexpected centromere phenotype that helps define the molecular basis for the pleiotropic effects of the kinetochore protein knockdown. Our data provide evidence for functional inter-dependence of the centromeres and the centrosome. These new findings highlight the usefulness of *P. falciparum* as a novel model system for investigating mitosis in eukaryotes.

We provide a point-by-point response to the reviewers' comments, below.

Reviewer #1. (Extracts of Reviewer's Comments shown in blue text)

The images shown in this article are beautiful and give a nice cellular overview of both the localisation of kinetochore proteins and their cellular requirement. However, the authors focus too much on the relationship between kinetochores and the apical complex blurring the main finding of this work. The link between the mitotic machinery and the apical complex in apicomplexans has been described, see for example PMID11085247 (2000) or PMID17604449 (2010) for a review. While I understand that implementation of expansion microscopy with NHS ester staining represents an accessible and powerful approach to visualise these structures, a similar description of the link between the mitotic machinery and the apical complex has already been performed comprehensively and well-reviewed by some of the co-authors (PMID 36993606 and 37571814) as was the relative position of kinetochore proteins with the centriolar plaque (PMID36006241).

Major general comments

While I understand that this represents an exciting and hot area of research, I think that by focusing on the link between the kinetochore and the apical complex the authors are avoiding to tackle the most relevant phenotype described in this article, ie, the destabilisation of the "nexus" between the apical complex and the mitotic spindle: the intervening MTOC. Non-specialist readers will likely be misled towards a functional link between the kinetochore and the apical complex, which is very indirect. In that respect the title is particularly misleading by highlighting the link between the "nexus", aka centriolar plaque, and the apical complex, which is already known.

I think the authors should recenter the wording of the paper on the stability of the centriolar plaque relative to the kinetochores and consolidate the phenotyping of the rupture of the centriolar plaque.

We thank the reviewer for their comment. "The images shown in this article are beautiful and give a nice cellular overview of both the localisation of kinetochore proteins and their cellular requirement."

We accept the reviewer's comment that similar descriptions of the link between the mitotic machinery and the apical complex and the relative position of kinetochore proteins with the centriolar plaque has already been published. The Reviewer suggests we should focus the paper on the "stability of the centriolar plaque relative to the kinetochores and consolidate the phenotyping of the rupture of the centriolar plaque".

To shift the focus, in the revised version of the manuscript (as described in detail below), we have:

- 1) Changed the title to “Disruption of *Plasmodium falciparum* kinetochore proteins destabilises the nexus between the centrosome equivalent and the mitotic apparatus”.
- 2) Moved the previous Figure 2, describing the link between the mitotic machinery and the apical complex, to Supplementary Data.
- 3) Provided additional data that describes the effects of NDC80/ Nuf2 cKD/KO on the microtubule organisation (Fig 2b, Supplementary Fig. 3b, top panels, Fig. 5a, d, first and second rows, Supplementary Fig. 11a, 13b and 14a) and centromere disruption (Fig. 1d, 5c, first and second rows, Supplementary Fig. 5a, b, top panels, Supplementary Fig. 10, first and second rows) in early rounds of nuclear division.
- 4) Included new data on the reorganisation of a centromere marker upon cKD/KO of *PfNUF2* and *PfNDC80* (Fig 1d, 3a, 5c, Supplementary Fig. 5, 10).

At a technical level – expansion microscopy has been instrumental to describe the effect of knockdown of kinetochore proteins in these cells. When coupled with immunolabelling of specific proteins, this technique opens the doorway to a medium throughput method at sub-EM resolution to phenotype mutants in a way that is not possible by EM. Instead, the authors focus on NHS ester and tubulin staining throughout the work, meaning the structures described would actually be seen at greater resolution by EM, and this would likely avoid confusion as to where each “continuous structure” delineates. Therefore, I am uncertain as to advantages here of using UExM compared to EM for their phenotyping. One example of this is the “dark spots” in the nucleus (as seen by NHS ester alone following UExM). How do you know these are not clusters of kinetochores? An anti-CENH3 may answer this – either stain these spots appropriately to gain the advantages of UExM or just use EM for greater resolution and indication of what they are composed of.

The reviewer makes the reasonable point that UExM does not provide sufficient resolution to enable us to determine the nature of the “dark spots” in the nucleus (as seen by NHS ester alone following UExM). The reviewer suggests labelling with a reagent recognising the centromere marker, *PfCenH3*.

We explored the available anti-*PfCenH3* antibody preparation but found that it did not provide convincing labelling in the samples prepared for U-ExM. We also tried using transmission EM as a means of obtaining information about the ultrastructure of these cellular feature. Densely stained MTOCs are evident in some control cells but not in cKO/KD cells. Densely stained amorphous features, that are separated from the nuclear periphery, are evident in some cKO/KD cells. These may be equivalent to the features observed by U-ExM. Nonetheless, we did not find these data sufficiently convincing.

Instead, we have generated two new cell lines, in which a plasmid encoding GFP-tagged CenH3 has been co-transfected into the *PfNDC80*-HA and *PfNuf2*-HA cKO/KD lines. In control cells, the GFP-CenH3 is observed close to the kinetochore marker, at all stages of the nuclear division process (Fig 1 d, Fig 3a, Supplementary Fig. 5, 10). Disruption of either of the kinetochore proteins, *PfNDC80* and *PfNuf2*, reveals a very interesting phenotype, leading to dispersal of the centromere marker, *PfCenH3*. These data suggest that the whole kinetochore/ centromere complex is destabilised. We have included the new data in new Fig 1d, 3a, b, 5c, Supplementary Fig. 5, 10).

The new data indicate that the dark foci in NHS-PBS-labelled samples are unlikely to be remnant kinetochore/ centromere complexes. Therefore, we have removed our statement speculating about the nature of the foci and instead indicated that further work is needed to establish the identity of these structures.

More specific comments

line 40 - change "substantive" to "substantial". Done

line 47 - change "divergent" to "different". Done

line 51 - what is "semi-synchronised"? We apologise for the confusion. There is some debate about the synchronicity of the final round of nuclear division, but cytokinesis is co-ordinated. We have changed the term to synchronous and added an additional reference (doi.org/10.1371/journal.ppat.1010595) that describes the analysis of synchronicity in detail.

Line 72 – Please also reference PMID36006241. Done.

Line 106 and supplementary Figure 1d legend - how was the 90% reduction of protein levels quantified to support "very efficient and sustained knockdown"? It is also not clear to what this is compared? For example, in *Toxoplasma gondii*, auxin-induced depletion often yields depletion to undetectable levels by western blotting and this is directly compared in discussion line 422. If this technology is to be compared, the data must support this claim. Please show a dilution series to support this 90% value. In addition, it is not indicated on which stages the blots are performed. Without timing depletions, it is not possible to interpret which mitosis is affected during schizogony or the possibility of any delays.

We have now included new PCR analysis and Western blot data at different time points post invasion (Supplementary Fig. 1 b-e). We accept the reviewers point that it is not possible to precisely measure the level of reduction in protein levels from the data provided. We have therefore removed the quantification statement from Supplementary Fig 1d legend. We have also removed the statement in the discussion that the difference compared with *Toxoplasma* may be due to less efficient knockdown. In view of our new findings with the centromere markers, CenH3, we have also changed this part of the Discussion to note that in the case of *Toxoplasma* " ..the daughter cell did not inherit chromatin and the centromere remained assembled and associated with the parent nucleus (Farrell and Gubbels, 2014).. "

Supplementary Figure 1a - primer binding sites and expected sizes are missing from schematic.

Primer binding sites and expected sizes have been provided. This section of Supplementary Fig 1b, c legend now reads: "The *ndc80*-glms product is 2425 bp and the excised *ndc80*-glms product is 791 bp. The *nuf2*-glms product is 2288 bp and the excised *nuf2*-glms product is 746 bp."

Supplementary Figure 1b - expected sizes are missing from gels.

Expected sizes have now been provided. This section of Supplementary Fig. 1d, e legend now reads: "*Pf*NDC80-HA (73 kDa) (d) and *Pf*Nuf2-HA (58 kDa) (e) proteins are detected by Western blotting (anti-HA) at the expected sizes at different time points, illustrating efficient cKO/KD, when compared to DMSO controls."

Supplementary Figure 1c - time post induction is missing from western blot. How long does it take to achieve this level of reduction?

We have now provided additional PCR-based analysis of the gene locus and Western blot analysis at different times post invasion (24 – 41 h) to provide a more detailed analysis of the knockdown efficiency at different stages (New Supplementary Fig. 1 b, c, d, e).

Supplementary Figure 1f legend - what is "conventional immunofluorescence microscopy".

"conventional" has been changed to "deconvolution" to make the method clearer.

Line 183 - "There is evidence for a connection from the rhoptries through to the NDC80-labelled inner centriolar plaque, as evidenced by a continuous strong NHS". I find this statement misleading. There are many structures between the rhoptries and kinetochores. These do not form single continuous structure. EM has previously clearly shown these structures, and also does in figure 4F.

We accept the reviewer's point and have removed the statement.

Lines 185 - "The NHS-PBS-labelled structure extends towards the PPM and appears to seed the formation of the inner membrane complex (IMC), apical complex and the cytoplasmic microtubules, in a process that appears to be driven by the positioning of the centriolar plaque (Fig. 2d)." Similarly, whilst Fig 2d shows a very nice image of the spindle and kinetochores (both NHS ester and immunostained), I do not believe this claim is supported. The overall structure is linked by multiple components, also including the "centriolar plaque" (a term coined by EM and not NHS ester staining).

And Line 199 – the IMC and the PP are not part of the apical complex, they form the pellicle.

We accept the reviewer's point. The first sentence has been removed and the 2nd changed to "Taken together with previous reports(Li et al., 2022; Liffner and Absalon, 2021), these data indicate connections from the spindle and inner centriolar plaque through to the outer centriolar plaque, the nascent cytoplasmic microtubules, the apical complex, and the IMC and the PPM."

Line 223 - "We investigated if PfNDC80 cKO/KD affects the final round of nuclear division (38-40 hpi) when the cell initiates cytokinesis and the nuclei are packaged into nascent merozoites". See earlier point regarding levels and timing. Without evidence of how long it takes for these proteins to be depleted by the cKO/KD method, it is not possible to know which round of nuclear division is affected. What if 50% protein levels are sufficient for mitosis in these cells and below 50% is only achieved after 35 hours? Line 231 and line 256 - The above comment would also support the partial KD fitness conferring phenotypes observed.

We have now included additional data (Supplementary Fig. 1b-e) that follows the knockdown of PfNDC80 and PfNuf2 (by PCR & Western) at different times post invasion (24 – 41 h). A substantial reduction in the PCR-based signal is observed from 24 hpi. In response to Reviewer 3, we have also now included some additional data looking at the knockdown phenotype in the earlier rounds of nuclear division. There appears to be a defect from the first round of nuclear division.

Line 296 "Inspection of the NHS-PBS-labelled features in PfNuf2-cKO/KD schizonts, reveals disruption of the centriolar plaque structure with a loss of the physical connection between the intra- and outer-nuclear structures". While this is a central statement of the paper I do not see this by these images and do not believe this can be shown by NHS ester staining as long as no specific markers of the outer and inner centriolar plaque are shown. Please show EM equivalents to support this claim. Line 398 - same as above, there is no clear evidence of the inner centriolar structure by NHS ester staining.

We accept the reviewer's point and now instead rely on the anti-centrin labelled samples to make the point that the centrin-containing outer centriolar plaque is physically separated from the nucleus upon cKD/KO. We also use the EM data to show that the apical complex is separated from the nucleus. As discussed above, our attempts to use EM to show that the inner centriolar plaque is no longer associated with the nuclear envelope proved difficult as it is hard to show the absence of a structure by EM. We have removed the statement from the Discussion that "the protein-dense inner centriolar structure appears to disengage from the luminal side of the nuclear membrane".

Line 387 - "Interestingly, disruption of PfNDC80 and PfNuf2 does not appear to halt chromatin duplication, karyokinesis or stall mitosis". Some of the most severe SAC mutants in human cells e.g. (PMID21664570) only delay entry into anaphase and do not provide a complete block. No experiment described in this paper have looked at the length of mitosis in these cells.

We accept the reviewer's point that we have not looked at the length of mitosis in the kinetochore-disrupted parasites. The live cell imaging of egressed parasites (Fig. 4e) was performed at about 41 hours post invasion. Many cells of both PfNDC80-disrupted and control parasite samples are able to egress. However, the timing and efficiency of the event were not monitored closely. Therefore, we have changed the wording to: "Interestingly, disruption of PfNDC80 and PfNuf2 does not appear to prevent chromatin duplication or karyokinesis, although there is evidence for uneven inheritance of chromatin in the divided nuclei."

Line 406 – The FBXO1 protein was previously shown to localise on between the forming roptries and the outer centriolar plaque and its depletion also led to aberrant formation of the apical complex and the IMC (PMID31348812 36898988).

We have now included reference to the SCF^{FBXO1} E3 ubiquitin ligase work.

Reviewer #2 (Remarks to the Author):

During the intraerythrocytic asexual stage, the Plasmodium parasites proliferate in a way called schizogony, in which several asynchronous rounds of endomitosis are followed by one round of karyokinesis and cytokinesis, forming a segmented schizont containing several daughter haploid nucleated merozoites. The mechanism underlying the precise separation and allocation of replicated chromosomes into daughter nucleus, the precise structure linking the nucleus and apical organelles are yet to be verified. The authors used ultrastructure-expansion microscopy and high-resolution imaging to explore the connection between nuclear spindle microtubules and cytoplasmic sub-pellicular microtubules during the asexual proliferation of Plasmodium falciparum. They found that the centriolar plaque, the chromosome-capturing structure kinetochore and apical complex are associated during the early stage of mitosis, functions as a nexus coordinating the formation of nuclear spindle and sub-pellicular microtubules. This is

the first study in the Plasmodium to reveal the relationship between the nuclear spindle and the cytoplasmic sub-pellicular microtubules in the asexual stage. Furthermore, the authors investigated the role of two kinetochore proteins NDC80 and Nuf2 during schizogony in Plasmodium falciparum. However, I have some comments and suggestions for the authors of this manuscript prior to its publication.

Major comments :

Line 59: "Aster fibres form on the other side of the centrosomes and radiate towards the poles of the dividing cell to position the spindle apparatus during mitosis." It is not described clearly to the reader.

The sentence has been removed.

Line 199-200: The conclusion is overstated and not supported by the experiment evidences. The NHS ester-conjugated, sulfonated fluorophores are water-soluble probes that unspecifically reacts with primary amines in proteins and other molecules to yield stable amide bonds. The imaging via this generalized dye can only tell that these structures (with dense signal) are associated, it is overstated that these structures are physically connected.

We accept the reviewers point. We accept that NHS labelling is not sufficient to provide evidence for a physical connection. However, we also note that Reviewers 1 & 3 suggest that there is already published data for a connection. To satisfy both points, we have changed the sentence to read: "Taken together with previous reports (Li et al., 2022; Liffner and Absalon, 2021), these data indicate a continuum of structures from the spindle and inner centriolar plaque through to the outer centriolar plaque, the nascent cytoplasmic microtubules, the apical complex, and the IMC and the PPM."

In the Figure 1d, the staining signal of the beta-tubulin (for the mitotic spindle) is not clear. In the current image, no clear microtubules (radiating form) could be seen. It is hard to judge the relative localization between Nuf2 protein and spindle microtubules. Suggest to replace with a better image.

The NHS-PBS data from Fig 1d has now been moved to Supplementary Fig. 3b; and a new panel has been provided. Fig. 1d now presents data for the centromere and kinetochore markers.

The Reviewer requests an image of "clear microtubules (radiating form)". It is important to note the dividing *P. falciparum* nucleus has a compact spindle. This is shown very clearly in the new data presented in Fig. 1c, d, 2b, 3a, Supplementary Fig. 5a, b, 10). The centromeres and kinetochores are co-located; and move to the mid-place during nuclear division. These data confirm that the kinetochore protein is located at a compact structure associated with the mid plane lying between the adjacent MTOCs. Our data are consistent with previous reports (Liffner et al., 2023).

The reviewer refers to a radiating form of the spindle. It is important to note that the extended spindle observed in *P. falciparum* dividing nuclei is not the equivalent of the mammalian anaphase spindle. This "radiating" form is observed after the chromatids have already been separated and the centriolar plaque/MTOC structures migrate away from each other. At this stage, the kinetochores relocate close to the nuclear periphery. Examples are evident in Fig. 1c (2nd row), Fig. 2b (2nd row), Fig. 3a (1st row, zoom 2), Fig. 5a (1st row), 5c (3rd row, zoom 1), Supplementary Fig. 2a, b, 3a, b, 5a, b (2nd panel).

In the Figure 1c and d, the kinetochore proteins NDC80 and Nuf2 were detected to be localized at the base site of spindle. No protein signal was seen along the spindle microtubule or at the plus end of microtubule. Based on the kinetochore localization compared to spindle, it is hard to speculate the roles of kinetochore in the separation of duplicated chromosomes. It is good to discuss this in the discussion section.

As discussed above, the kinetochore proteins *PfNDC80* and *PfNuf2* are observed at the mid-plane in the division apparatus that forms after the centriolar plaques/ MTOCs duplicate. We have now provided new data on *PfNDC80/PfNuf2*-HA/GFP-*PfCenH3* co-transfectants (Fig. 1d, 3a, 5c, Supplementary Fig. 5a, b, 10) that nicely illustrates the reorganisation. We have rewritten a section of the Discussion to further emphasise this point.

Line 357 " Following duplication of the centriolar plaque structures, a short mitotic spindle is established and *PfNDC80*, *PfNuf2* and *PfCENH3* relocate to the mid-plane of the spindle, consistent with the role of segregating sister

chromatids (Brusini et al., 2022; Liffner et al., 2023; Zeeshan et al., 2021). The NHS-PBS-labelled region also moves along the spindle. The two centriolar plaque structures then migrate away from each other, and an extended spindle is observed (Liffner et al., 2023). It is important to note that this is not the equivalent of the mammalian anaphase spindle as the chromatids have already been separated. Next, the kinetochore and centromere markers retract back towards the region below the centrin puncta, and the NHS-PBS-labelled inner centriolar plaque structure is again evident.”

Fig 2b: PBS-NHS-labelled structure (Centrin is associated with a punctate NHS-PBS-labelled structure just outside the nucleus) is not clear.

We have changed the text to read “The centrin-labelled punctum is located on the cytoplasmic side of the nuclear membrane”.

The kinetochore is not functional, how the chromosomes could be separated after duplication? In the Figure 3, 4, and 5, both the NDC80 cKO/KD and Nuf2 cKO/KD parasites displayed defected spindle microtubules inside the nucleus, decoupled connection between spindle and sub-pellicular microtubules, and daughter merozoite without nucleus. However, in the schizonts of the mutant parasites (NDC80 cKO/KD and Nuf2 cKO/KD) shown in these figures, most of the nucleus were successfully separated after chromosome duplication. Have the authors observe any schizont containing a big duplicated nucleus without no separation.

The Reviewer raises a point that greatly intrigues us. How can the daughter chromatids separate and be divided among the daughter nuclei when the connection to the spindle microtubules is lost. Of interest, disruption of α 1-tubulin has been reported to cause aberrant microtubules formation but not affect the genome duplication and nuclear division in *P. berghei* oocysts (Spreng et al., 2019). As presented in Supplementary Fig. 12f, there is evidence for uneven inheritance of chromatin in the divided nuclei. Quantification reveals a significant decrease in the mean volume of the (expanded) nuclei of cKO/KD parasites ($37 \pm 34 \mu\text{m}^3$) compared with controls ($43 \pm 10 \mu\text{m}^3$), as well as the presence of some very large volume nuclei (up to $200 \mu\text{m}^3$), consistent with uneven inheritance of the nuclear contents.

Minor comments :

Line 62: The authors mention that “By contrast, *P. falciparum* does not have centrioles, nor any other distinct structure within the centrosome equivalent, which is termed the centriolar plaque.” However, throughout the manuscript, inner centriolar plaque and out centriolar plaque were shown several times. It is better to say “no canonical centrioles in Plasmodium”

The sentence has been changed to: “..*P. falciparum* has no canonical centrioles.”

FigS1d-e: For the 90% decrease in the expression of both NDC80 protein and Nuf2 protein described in the figure legend, a quantitation analysis should be added in the figure.

As described above, in response to Reviewer 1, we accept that it is not possible to quantitate the level of reduction in protein levels without a more detailed analysis. We have therefore removed the quantification statement from Supplementary Fig. 1d legend, and instead provided additional data (Supplementary Fig. 1b-e) to enable a qualitative assessment of *PfNDC80* and *PfNuf2* at both the gene and protein level at different times after treatment.

FigS1d-e: As ERC is an internal reference protein, the description of this internal reference protein needs to be added.

We repeated the Western blotting experiments to include samples at different times post invasion. We replaced all the Western blots. We used a new loading control protein, anti-*Pfaldolase* (Hollin et al., 2022) and have removed reference to *PfERC*.

Line 120: A citation should be provided for “the protein-rich structures, such as the rhoptries”.

A recent reference which provides an atlas of NHS-labelled structures in *P. falciparum* (Liffner et al., 2023) has now been provided in support of this statements.

Line 275: “...positioned at one end of the merozoite and the nucleus occupying the basal region”. Rephrase this sentence for good understanding.

This sentence has been rephrased: “The electron dense apical organelles, *e.g.*, rhoptries (yellow arrows), are positioned at one end of the merozoite, while the nucleus is observed in the basal region”

Reviewer #3 (Remarks to the Author):

This study provides enticing insight into the positioning and function of the outer kinetochore proteins of the malaria-causing parasite *Plasmodium falciparum*. While the very first sections of the manuscript reiterate some previously describe organization principles of the mitotic machinery the authors uncover a highly intriguing phenotype resulting from kinetochore protein depletion. They note a loss of centriolar plaque integrity resulting in spindle misorganization and detachment of the nuclei from their plasma membrane-localized budding sites. While the presented findings are specific to the apicomplexan field this work highlights the divergence of eukaryotic cell division biology and how canonical components are differentially used by malaria parasites to drive their atypical nuclear multiplication and segregation. The authors make good use of the now well-established ultrastructure expansion microscopy protocol while introducing a small variation that strongly accentuates the visibility of membranes, including the nuclear membrane. They also implement less commonly used array tomography to reveal more phenotypical details in their mutants. The presented study is coherent, and the conclusions and claims made in the manuscript are supported by the data. Yet some additional analysis might be required to clarify the mechanistic pictures the authors are drawing about the roles of kinetochores in centriolar plaque integrity, which I will outline below.

We thank the Reviewer for their generally positive comments.

Major comments

1) In the figures 1 and 2 the authors provide new details about the positioning of kinetochores within dividing nuclei of asexual blood stage parasites. They also reiterate several findings that have been made previously concerning kinetochore dynamics (Brusini et al.), inner and outer centriolar plaque structure (Liffner et al. and Simon et al.), and microtubule organization (Balestra et al.). Yet they repeatedly use a wording implicitly suggesting that their study is the first to show this as in line 132ff, but more specifically line 199ff with statements like “...these data indicate a physical connection from the spindle and inner centriolar plaque...” and “data further suggest that the spindle microtubules nucleate from a protein-dense structure...”. Both findings having been described previously.

We apologise for giving this impression as we had tried to be careful to make it clear that the data confirm and build on previous studies. Indeed, our manuscript includes authors from at least one of the studies referred to.

We note that this is the first study in *P. falciparum* asexual stage transfectants in which of the kinetochore proteins have been labelled and expansion microscopy used to assess the dynamics of the interaction. So, our data on the dynamics of kinetochore proteins in *P. falciparum* asexual stages is novel (for this parasite species). Nonetheless, we are very aware that the kinetochore protein dynamics have been studied in *P. berghei* (Brusini et al., 2022; Zeeshan et al., 2021). And of course, microtubule and centriolar plaque organisation has been studied by different groups. Our work is highly consistent with these previous studies, and we are keen to make that clear. The data are presented here as a stepping stone to our presentation of the data for the *PfNDC80/PfNuf2* knockdowns.

The text has been changed and now reads: The dynamic organisation of kinetochore proteins, Nuf2, NDC80 and Apicomplexan Kinetochore protein1 (AKiT1), has previously been studied in *P. berghei* (Brusini et al., 2022; Zeeshan et al., 2021). Consistent with those reports, in nuclei preparing for karyokinesis, *PfNDC80* and *PfNuf2* signals (Fig. 1c, Supplementary Fig. 2a, b, 3a, b, 1st row, yellow arrowheads) are concentrated near the base of anti- β -tubulin-labelled hemispindles, which are located at the periphery of the nucleus, as marked by DAPI (blue; Fig. 1c, Supplementary Fig. 2a, b, 3a, b) and by the NHS-BSA-labelled membrane (green), as reported previously (Gerald et al., 2011; Liffner and Absalon, 2022; Read et al., 1993). During nuclear division, *PfNDC80* and *PfNuf2* (Fig. 1c, Supplementary Fig. 2a, b, 3a, b, 2nd row, yellow arrowheads) are often observed as a pair of parallel lines at the mid-plane of the mitotic spindle.”

Similarly in the discussion (lines 336 ff) the authors use the wording “We showed...” going on to describe previously published data. Even though the authors conclude the respective result paragraphs (lines 142 & 205) with the mention that their data is consistent with previous publication this does not clarify sufficiently to a non-expert reader the degree of novelty of their contribution. I understand that it is important to reiterate those findings for comprehensiveness, but strongly suggest rewording of the aforementioned sections accordingly using words like confirm / validate.

This section has been rewritten.

Line 346. “Consistent with previous reports(Brusini et al., 2022; Li et al., 2022; Zeeshan et al., 2021), an early event is the production of a hemispindle with about five bundles of microtubules, emanating from a nuclear lumen-facing protein-dense (NHS-PBS-labelled) structure(Brusini et al., 2022; Li et al., 2022; Zeeshan et al., 2021).”

Line 367 “The apparent plasticity of kinetochore organisation is consistent with previous live cell imaging studies in *P. berghei* and *P. falciparum*, as well as in other eukaryotic cells (Brusini et al., 2022; Li et al., 2022; Zeeshan et al., 2021).”

2) A potential discrepancy between the data shown in this manuscript (Fig. 1 & 2) and previous findings is the positioning of the kinetochore signal in the hemispindle stage. The authors show a positioning within the inner centriolar plaque, which has been previously described as chromatin-free. Further CenH3, the centromeric histone onto which kinetochore components bind, has been shown to localize more on the edge of the inner centriolar plaque (Simon et al.). One way to resolve this discrepancy is that potentially some of Ndc80 is not chromatin bound or maybe that the kinetochore as a whole significantly extends into the protein dense region. A key difference to previous publication is that this study focusses more on later stages of nuclear division. The authors should comment on this, but more importantly show a few more images of control cells and mutants of early schizont stages with 1-3 nuclei i.e. during their first nuclear divisions. This is relevant a) to resolve the aforementioned discrepancy, b) since kinetochores are likely already required for the first nuclear division it is critical to analyze a potential early phenotype to clarify whether a later phenotype is a just an “escalation” of previous failures to e.g. segregate chromosomes c) Early and late schizont stages are differently organized, i.e. by presence/absence of rhoptries and subpellicular microtubules (Liffner et al), nuclear size an positioning, and should therefore be investigated separately.

We thank the Reviewer for this question. We don't see a discrepancy between our data and that of Simon *et al.* Those authors described an intranuclear region close to the membrane embedded centriolar plaque that exhibits reduced DNA staining and high protein staining, associated with microtubules. That is equivalent to our finding for the hemi-spindle, extended spindle & remnant spindle stages. When the mitotic spindle in formed between duplicated MTOCs, the NHS-PBS-labelled material relocates to mid-plane. Our interpretation is that the kinetochores are concentrated at the region of the inner centriolar plaque (and likely contribute to the observed density). While the chromatin is presumably attached to the kinetochores during the whole process, it seems that the bulk of the chromatin lies outside the protein-rich area.

In an effort to further address this question, we have generated two new cell lines, in which a plasmid encoding GFP-tagged *PfCenH3* has been co-transfected into the *PfNDC80* and *PfNUF2* inducible knock-down lines. In control cells the GFP-*PfCenH3* is observed close to the kinetochore marker, as expected.

The experiments in the kinetochore proteins are disrupted revealed a very interesting phenotype, with evident dispersal of the centromere marker, *PfCenH3*. We have now included the data for the *PfNDC80* and *PfNUF2* inducible knock-down/ GFP-*PfCenH3* lines in Fig. 3a, 5c, Supplementary Fig. 10.

We have also collected additional data in early rounds of nuclear division. As presented in Fig. 1d (1st and 2nd rows), Fig. 2b, Fig. 5a, c, d (1st and 2nd rows), Supplementary Fig. 3b (1st panel), 5a, b (1st and 2nd rows), 10 (1st and 2nd panels), 11a, 13b, 14a, the *PfNDC80* and *PfNUF2* markers are evident as puncta close to the nuclear periphery in untreated cells. Upon treatment with RAPA/ GlcN, substantive distortion of the nuclear envelope is observed, suggesting that the phenotype is evident from the early rounds of nuclear division (Fig. 2b, (4th, 5th, 6th panels), Fig. 5a, c, d, (2nd panel), Supplementary Fig. 10, 11 a, 13b, 14a (2nd panel).

3) An aspect of this intriguing study that needs clarification is whether the parasites experience a chromosome segregation defect due to the well described function of outer kinetochore proteins in this context. Visually Fig. 3c suggest that there might indeed be some errors in partitioning DNA, which is coherent with the increased nuclear volume observed in a subset of nuclei (Fig. S8). Yet is unclear to me how this can be reconciled with the equal final amount of nuclei observed while the number of formed merozoites (including empty merozoites) is lower in the mutant. I would not go as far as suggesting FISH analysis to quantify chromosome segregation errors, but maybe the authors can attempt a more thorough quantification of absolute DNA content in postmitotic nuclei (preferably using early schizont stages) to address this.

We agree with the Reviewer that this is an interesting question. The observation of the increased nuclear volume (in a subset of nuclei) and decreased nuclear volume in the majority of nuclei is, to our mind, not inconsistent with the equal final number of nuclei observed. Some nuclei inherit more DNA, some less, without a change in the total number of nuclei.

The moderate (20%) decrease in the number of nascent merozoites/ apical complex structures (and consequently the ratio of merozoites to nuclei) in the knockout suggests that there is a partial defect in formation of these complexes. However further work is required to understand the molecular basis of this defect.

As discussed above, we have collected data for early rounds of nuclear division and observe deformed nuclei that may underpin the defect in division process. However, we feel that a detailed quantification of chromosome segregation errors is beyond the scope of this manuscript. We do not feel comfortable trying to quantify the absolute DNA content in post-mitotic nuclei in UxEm-treated samples. As the Reviewer points out, a robust analysis would require a FISH study to determine whether different gene loci are correctly segregated. We agree with the reviewer that this is beyond the scope of the current manuscript. The generation of the knockdown parasites described in this work provides an important tool to address such questions in the future.

4) In Fig. 5a the authors provide, an interesting NHS-PBS staining for the Nuf2 knock down. It shows a protein dense region, supposedly the inner CP, which is detached from the nuclear periphery while still being associated with polymerizing microtubules. This is very intriguing and a critical piece of evidence that the kinetochore “nexus” is essential for CP integrity. Therefore, the authors should provide the same analysis for the Ndc80 knock down. Consequently, I feel there is a discrepancy between model Fig. 6 and Fig. 5a where NHS-PBS staining suggest that the detached inner CP is still the site of microtubule nucleation, while in the model these are not draw as overlapping. Please clarify the model accordingly once evidence for the Ndc80 KD has been added.

We thank the Reviewer for this question. As described above, we have now analysed a GFP-*PfCenH3*/ *PfNDC80* cKO/KD cell line and GFP-*PfCenH3*/ *PfNUF2* cKO/KD cell line. Upon kinetochore protein knockdown, the *PfCenH3* marker is dispersed (new Fig. 3a, b, 5c and Supplementary Fig. 10). In these cells, we still observed NHS-labelled structures in the nuclei, but it is now clear that they are unlikely to be remnant kinetochore/ centromere structures. We have therefore modified the text to note that the nature of the NHS-bright structures in the nucleus remains unknown. We have also modified the model to illustrate the loss of inner centriolar plaque structure upon knockdown but without speculating on the molecular detail.

Minor comments

5) Line 134: Please change the term “interpolar spindles” to “anaphase spindle” (or “extended spindle”). The term interpolar refers to a subset of microtubules within a spindle, which fully connect the two poles (hence interpolar), not to the spindle as a whole (which contains several “categories” of microtubules).

“interpolar” has been changed to “extended”.

6) Please briefly quantify the spindle microtubule phenotype observed in Fig. 3b. A mention of how many of the total imaged nuclei/cells display this phenotype in the text would suffice.

The previous Fig. 3b is now Fig. 2c. We have also added data for NHS-PBS labelling as Fig. 2b. We have now provided an analysis as Fig. 2d. The phenotype is evident in 100% of cells analysed.

Fig. 2d Analysis of the percentage of normal spindle microtubules in anti-HA-positive control and anti-HA-negative cKO/KD cells. Control, n = 20; cKO/KD, n = 20. The mean and standard deviation are plotted. Individual data points are shown. Statistical differences were determined using an unpaired Mann-Whitney t-test (****p < 0.0001).

References

- Brusini, L., N. Dos Santos Pacheco, E.C. Tromer, D. Soldati-Favre, and M. Brochet. 2022. Composition and organization of kinetochores show plasticity in apicomplexan chromosome segregation. *J Cell Biol.* 221.
- Farrell, M., and M.J. Gubbels. 2014. The *Toxoplasma gondii* kinetochore is required for centrosome association with the centrocone (spindle pole). *Cell Microbiol.* 16:78-94.
- Gerald, N., B. Mahajan, and S. Kumar. 2011. Mitosis in the human malaria parasite *Plasmodium falciparum*. *Eukaryot Cell.* 10:474-482.
- Hollin, T., S. Abel, A. Falla, C.F.A. Pasaje, A. Bhatia, M. Hur, J.S. Kirkwood, A. Saraf, J. Prudhomme, A. De Souza, L. Florens, J.C. Niles, and K.G. Le Roch. 2022. Functional genomics of RAP proteins and their role in mitoribosome regulation in *Plasmodium falciparum*. *Nature Communications.* 13:1275.
- Li, J., G.J. Shami, E. Cho, B. Liu, E. Hanssen, M.W.A. Dixon, and L. Tilley. 2022. Repurposing the mitotic machinery to drive cellular elongation and chromatin reorganisation in *Plasmodium falciparum* gametocytes. *Nat Commun.* 13:5054.
- Liffner, B., and S. Absalon. 2021. Expansion microscopy reveals *Plasmodium falciparum* blood-stage parasites undergo anaphase with a chromatin bridge in the absence of mini-chromosome maintenance complex binding protein. *Microorganisms.* 9:2306.
- Liffner, B., and S. Absalon. 2022. Hand-in-hand advances in microscopy and *Plasmodium* nuclear biology. *Trends Parasitol.* 38:421-423.
- Liffner, B., A.K. Cepeda Diaz, J. Blauwkamp, D. Anaguano, S. Frölich, V. Muralidharan, D.W. Wilson, J. Dvorin, and S. Absalon. 2023. Atlas of *Plasmodium falciparum* intraerythrocytic development using expansion microscopy. eLife Sciences Publications, Ltd.
- Read, M., T. Sherwin, S.P. Holloway, K. Gull, and J.E. Hyde. 1993. Microtubular organization visualized by immunofluorescence microscopy during erythrocytic schizogony in *Plasmodium falciparum* and investigation of post-translational modifications of parasite tubulin. *Parasitology.* 106 (Pt 3):223-232.
- Spreng, B., H. Fleckenstein, P. Kubler, C. Di Biagio, M. Benz, P. Patra, U.S. Schwarz, M. Cyrklaff, and F. Frischknecht. 2019. Microtubule number and length determine cellular shape and function in *Plasmodium*. *EMBO J.* 38:e100984.
- Zeeshan, M., R. Pandey, D.J. Ferguson, E.C. Tromer, R. Markus, S. Abel, D. Brady, E. Daniel, R. Limenitakis, and A.R. Bottrill. 2021. Real-time dynamics of *Plasmodium* NDC80 reveals unusual modes of chromosome segregation during parasite proliferation. *Journal of cell science.* 134:jcs245753.

REVIEWERS' COMMENTS

Reviewer #1 (Remarks to the Author):

The authors have nicely taken into account our comments in this revised version. I would like to congratulate them on this beautiful piece of work.

If I am not mistaken, they however do not describe in the method section the details about the generation of the CENH3 expression vector.

Reviewer #2 (Remarks to the Author):

The authors have addressed the major issues. However, I have two minor comments for the authors of this manuscript.

Line 236 and Figure 3a: After the PfNDC80-HA cKO/KD, no NDC80 signal was detected in the nucleus compared to the control group. The authors stated that "PfCENH3 disperses throughout the nucleus". Another possibility is that PfCENH3 may be unstable and degraded in the nucleus.

In the manuscript, the author uses "Supplementary Fig. 6a" (line 174) and "Suppl Fig. 6a" (line 173), please use the same format throughout the manuscript.

Reviewer #3 (Remarks to the Author):

The authors have made a great effort to address the raised reviewer comments. The text has been appropriately edited to acknowledge more clearly previous findings. Showing early division stages was very helpful to clarify that the phenotype is indeed resulting from issues in centriolar plaque integrity from the beginning and not a consequence of multiple rounds of failed chromosome segregation events. The additional CenH3 localization data is coherent, informative, and clarifies that the intranuclear region of the centriolar plaque is indeed harboring kinetochores as well as centromeres. I have no further major comments.

Minor comment:

Line 247: Remove „-HA“ from PfNuf2-HA.

AUTHOR'S RESPONSE TO REVIEWERS

Reviewer #1.

We have expanded the method section to include the details about the generation of the *Pf*CENH3 expression vector.

Reviewer #2.

We have modified the discussion to include the possibility that *Pf*CENH3 may be unstable and degraded.

We have made the other minor suggested changes.

We have modified the figure legends to include relevant the statistics data.